# R3: Robust Rubric-Agnostic Reward Models

### Abstract

Reward models are essential for aligning language model outputs with human preferences, yet existing approaches often lack both controllability and interpretability. These models are typically optimized for narrow objectives, limiting their generalizability to broader downstream tasks. Moreover, their scalar outputs are difficult to interpret without contextual reasoning. To address these limitations, we introduce R3, a novel reward modeling framework that is rubric-agnostic, generalizable across evaluation dimensions, and provides interpretable, reasoned score assignments. R3 enables more transparent and flexible evaluation of language models, supporting robust alignment with diverse human values and use cases. Our models, data, and code are available as open source.

## 1 Introduction

Reward models play a central role in aligning language model outputs with human preferences by assigning scalar scores to generated responses (Rafailov et al., 2023; Lambert et al., 2024b). However, current reward modeling approaches suffer from two significant limitations: limited controllability and poor interpretability. First, these models are often optimized for narrow objectives—such as helpfulness or harmlessness—resulting in behavior that is overly tailored to specific metrics and not readily generalizable to a broader range of downstream tasks (Li et al., 2019; Stureborg et al., 2024). Second, the interpretability of reward scores remains unclear. For instance, scalar values like "1" or "2" on a Likert scale are not inherently meaningful without an explicit explanation of what those scores represent in context.

Aligning models with human preferences is crucial, but obtaining human judgments is often costly and time-consuming (Vu et al., 2024; Lin et al., 2025; Winata et al., 2025). Leveraging existing human evaluations from prior research appears promising; however, it poses several challenges, including lack of standardization, varying evaluation criteria, insufficient documentation, data privacy issues, and proprietary restrictions (Kim et al., 2024a). As an alternative, using model-generated outputs for reward modeling or annotation offers greater efficiency and flexibility. This lack of generalizability and transparency presents challenges for reliably evaluating and guiding language model behavior across diverse use cases. To address these issues, we propose **R3**, a novel reward modeling framework that is rubric-agnostic, generalizable to various evaluation dimensions, and grounded in interpretable, measurable scores. Our approach not only supports more flexible alignment with human values but also includes explicit reasoning for score assignments, enabling more transparent and trustworthy model evaluation. Our contributions can be summarized as follows:

- We introduce R3, a novel task-agnostic robust reward model training framework that leverages fine-grained rubrics to provide highly controllable and interpretable reward scores. These rubrics can be either hand-crafted by humans or generated by LLMs.

- We propose a unified framework for training reward models by adapting various types of data into three standard formats: point-wise, pair-wise, and binary.

- We curate a new reward modeling R3 dataset collected from 45 diverse sources that covers tasks such as classification, preference optimization, and question answering (Figure 2). Each example in the dataset contains an instruction and task description, input, response(s), evaluation rubrics, and a score along with the corresponding reasoning (Figure 1).

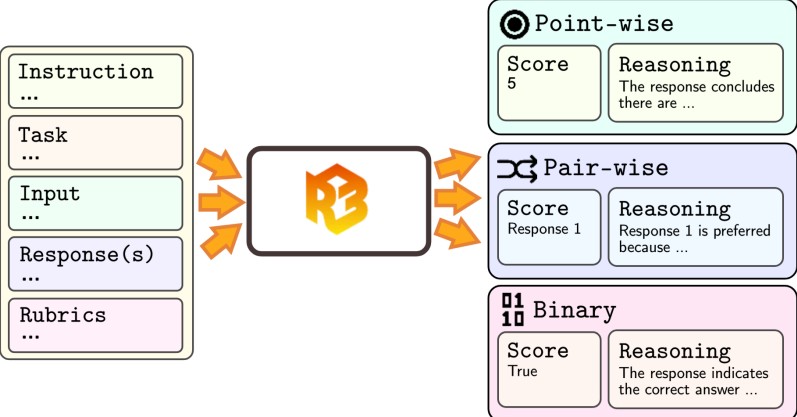

Figure 1: ROBUST RUBRIC-AGNOSTIC REWARD (R3) models both the input and output of a task. It takes a prompt that includes an instruction, task description, input, response(s), and evaluation rubrics, and generates a score along with the corresponding reasoning.

- We demonstrate that our R3 models exhibit robust and superior performance, not only matching but often exceeding both established baselines and proprietary models across a diverse suite of tasks, including reward modeling, knowledge recall, reasoning, and summarization. Importantly, our framework maintains this high level of effectiveness even under stringent resource constraints—utilizing no more than 14,000 training examples and limited computational capacity—by leveraging efficient adaptation techniques such as low-rank adaptation (LoRA) (Hu et al., 2022).

## 2 Aren't Existing Reward Models Robust Enough?

The challenge of building models that generalize across diverse tasks and domains—particularly in evaluating quality from multiple aspects or human annotation metrics—is well established. In this section, we present the motivation behind the need for developing new reward models.

**Controllability.** Existing reward models, such as ArmoRM (Wang et al., 2024a) and UniEval (Zhong et al., 2022), offer limited support for evaluating models on fine-grained aspects. They typically require separate training for each aspect along with corresponding parameter weights, reducing flexibility during both training and evaluation—especially when dealing with unseen aspects. Similarly, models like Prometheus (Kim et al., 2023; 2024b) are restricted in the range of supported task types; for example, they do not accommodate binary classification. ArmoRM is further limited in that it only supports point-wise tasks, making it unsuitable for pair-wise comparisons.

**Interpretability.** Scores generated by reward models—particularly those based on generative LLMs (Shiwen et al., 2024; Yu et al., 2025) or custom classifiers (Wang et al., 2025a; Winata et al., 2024; Zhang et al., 2024c) —can be difficult to interpret. For example, a score of 0.6543 on a 0–1 scale offers little clarity: Is it measuring helpfulness, correctness, coherence, or some opaque combination of all three? Without a clearly defined rubric or accompanying explanation, such scores provide limited actionable insight, leaving users to guess what aspect of quality the number is intended to capture.

**Limited Compatibility on Various Tasks.** Existing reward models often have limited compatibility with a diverse range of tasks. For instance, models like RM-R1 (Chen et al., 2025b) are primarily designed for pair-wise comparisons, making them less suitable for point-wise or binary classification tasks, which limits their applicability. Similarly, Prometheus supports point-wise and pair-wise evaluations but lacks native support for binary classification—an approach that can be particularly effective for tasks like hallucination or toxicity detection, where simple binary judgments are often sufficient for assessing data quality.

Table 1: A comparison between existing models and R3 across various dimensions, including data types, task formats, and evaluation rubrics. *The model is neither closed-source nor proprietary.

| Method | Data | | | | Tasks | | | Rubrics | Access* |
|---|---|---|---|---|---|---|---|---|---|
| | Size | Point-wise | Pair-wise | Binary | Point-wise | Pair-wise | Binary | Customizable | |
| ArmoRM (Wang et al., 2024a) | ∼974.4k | ✓ | ✓ | - | ✓ | - | - | - | ✓ |
| CLoud (Ankner et al., 2024) | ∼280k | ✓ | - | - | ✓ | - | - | - | ✓ |
| GenRM (Zhang et al., 2024a) | ∼157.2k | ✓ | - | ✓ | ✓ | - | ✓ | - | - |
| JudgeLRM (Chen et al., 2025a) | 100K | ✓ | - | - | ✓ | ✓ | - | ✓ | ✓ |
| Prometheus1 (Kim et al., 2023) | 100k | ✓ | - | - | ✓ | ✓ | - | ✓ | ✓ |
| Prometheus2 (Kim et al., 2024b) | 300k | ✓ | ✓ | - | ✓ | ✓ | - | ✓ | ✓ |
| m-Prometheus (Pombal et al., 2025) | 480k | ✓ | ✓ | - | ✓ | ✓ | - | ✓ | ✓ |
| Self-Taught (Wang et al., 2024c) | ? | - | ✓ | - | - | ✓ | - | ✓ | ✓ |
| SynRM (Ye et al., 2024) | 5k | ✓ | ✓ | - | - | ✓ | - | - | - |
| UniEval (Zhong et al., 2022) | ∼185.5k | - | - | ✓ | - | - | ✓ | ✓ | ✓ |
| G-Eval (Liu et al., 2023) | ? | ? | ? | ? | ✓ | ✓ | ✓ | ✓ | - |
| FLAMe (Vu et al., 2024) | 5M+ | ✓ | ✓ | ✓ | ✓ | ✓ | ✓ | ✓ | - |
| RM-R1 (Chen et al., 2025b) | ∼100k | - | ✓ | - | - | ✓ | - | ✓ | ✓ |
| **R3** | {4k, 14k} | ✓ | ✓ | ✓ | ✓ | ✓ | ✓ | ✓ | ✓ |

## 3 Tasks and Datasets

The goal of our open-ended evaluation model is to assess the quality of a response according to human-defined criteria, producing both a final score and a natural language explanation for interpretability. Formally, given a task instruction $t$, input instance $i$, one or more candidate responses $a$, and an evaluation rubric $r$, the model is tasked with generating an explanation $e$, that justifies the evaluation and a score $s$ that reflects the response quality under the given rubric $r$. We define this evaluation process as a function:

$$f(x) = y, \quad \text{where } x = (t, i, a, r) \text{ and } y = (e, s). \tag{1}$$

### 3.1 Task Formats

To support a wide range of evaluation settings, we define three task formats within our unified framework: *point-wise*, *pair-wise*, and *binary* evaluation. Each format shares the same input structure $x = (t, i, a, r)$ and output structure $y = (e, s)$ but differs in how the candidate responses are structured and how the score $s$ is defined.

**Point-wise Evaluation.** This format assesses the quality of a single response $a_1$ by assigning an integer score, typically on a 1–5 scale (Kim et al., 2023). It is suitable for open-ended generation tasks where scalar assessments of quality are needed, such as helpfulness, relevance, coherence, etc. Formally,

$$a = a_1, \quad f_{point-wise}(t, i, a, r) = (e, s), \quad s \in \{1, 2, 3, 4, 5\}. \tag{2}$$

**Pair-wise Evaluation.** In this setting, the model compares two candidate responses $a_1$ and $a_2$ to the same input $i$ and selects the preferred one, along with an explanation. This format is commonly used in preference-based training. Formally,

$$a = (a_1, a_2), \quad f_{pair-wise}(t, i, a, r) = (e, s), \quad s \in \{a_1, a_2\}. \tag{3}$$

**Binary Evaluation.** Binary task requires the model to make a definitive judgment about the correctness or acceptability of a response $a_1$, given the input and rubric. These tasks span a variety of use cases, including factual verification, binary classification (e.g., determining whether a summary is faithful), and structured reasoning (e.g., assessing the validity of a math or code solution). Formally,

$$a = a_1, \quad f_{binary}(t, i, a, r) = (e, s), \quad s \in \{\texttt{true}, \texttt{false}\}. \tag{4}$$

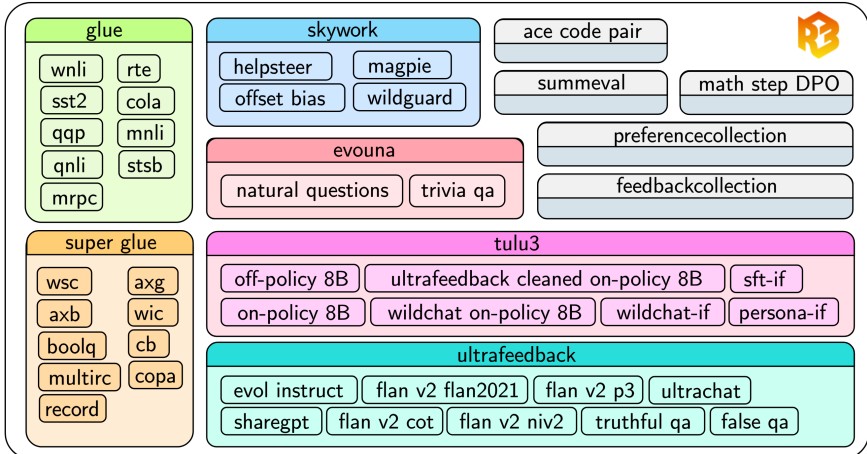

Figure 2: Dataset sources utilized in training the R3 model.

## 3.2 R3 Datasets

To support open-ended evaluation across diverse domains and task formats, we begin with a large pool of publicly available datasets spanning over 4 million examples, which include general chat, reasoning, and classification tasks, as shown in Figure 2. However, most of these datasets lack consistent evaluation rubrics and explanation traces, which are key components to train our evaluation model to output both scores and natural language justifications. Generating such traces, particularly using a strong reasoning model such as DeepSeek-R1 (Guo et al., 2025), is also computationally expensive and infeasible on a large scale.

To address this, we build our training dataset in multiple stages, drawing inspiration from Muennighoff et al. (2025) to emphasize both quality and diversity of the training data while on a limited budget. We first sample a diverse subset from the raw pool, then enrich each example with on-the-fly rubric generation and explanation traces. Finally, we apply filtering and refinement to produce smaller, higher-quality datasets used in supervised training. The following sections describe each stage.

### 3.2.1 Initial Curation

We begin by curating a large collection of publicly available datasets, denoted by $\mathcal{D}_{init}$, which spans on three broad categories: general chat, reasoning, and classification or evaluation tasks. Each example $x^{(j)} \in \mathcal{D}_{\text{init}}$ is a tuple $x^{(j)} = \left(t^{(j)}, i^{(j)}, a^{(j)}, r_{opt}^{(j)}\right)$, where $r_{opt}^{(j)}$ is optional rubric from the original dataset.

- **General Chat and Instruction-Following:** This category includes open-domain instruction tuning and user preference data, drawn from resources such as the Tulu subset (Lambert et al., 2024a), UltraFeedback (Cui et al., 2023), and Skywork Reward Preference (Liu et al., 2024a). These datasets contain point-wise and pair-wise tasks.

- **Reasoning Tasks:** To support math and code reasoning evaluations, we include datasets like Math-Step-DPO-10K (Lai et al., 2024) and AceCodePair-300K (Zeng et al., 2025), which contain preference annotations focused on correctness and reasoning quality on math and coding tasks.

- **Classification and Factual Evaluation:** This category consists of binary and pair-wise tasks aimed at assessing factuality, consistency, and alignment with task rubrics. We include GLUE (Wang et al., 2018), SuperGLUE (Wang et al., 2019), SummEval (Fabbri et al., 2021), FeedbackCollection (Kim et al., 2023), PreferenceCollection (Kim et al., 2024b), and EVOUNA (Wang et al., 2023). These tasks span summarization, natural language inference, general rubric-based classification, and factual correctness.

To construct binary-labeled data that includes *false* scores, we need to generate negative answers, as many datasets only provide the correct response (e.g., EVOUNA, GLUE, SuperGLUE). When possible, we sample negative answers from existing multiple-choice options. Otherwise, we generate negative answers using GPT-4.1 mini (Achiam et al., 2023).

### 3.2.2 Diversity Sampling

To ensure feasibility for distilling reasoning traces while maintaining representative coverage across domains and reducing redundancy, we downsample $\mathcal{D}_{\text{init}}$ to a 20k-example subset $\mathcal{D}_{20k} \subset \mathcal{D}_{\text{init}}$, manually allocating quotas per dataset to balance task types and formats. For each dataset in $\mathcal{D}_{init}$, we perform a three-stage sampling process to extract the most diverse examples:

1. **Embedding and Preprocessing.** Each instance is represented semantically by combining its instruction and input, $h(x^{(j)}) = t^{(j)} \oplus i^{(j)}$ and computing embeddings $Emb(h(x^{(j)})) = q^{(j)}$.

2. **Cluster Determination and Assignment.** Samples are grouped into clusters, with the number of clusters $k^*$ chosen to maximize the average Silhouette score:

$$k^* = \arg\max_k \frac{1}{|\mathcal{D}|} \sum_{j=1}^{|\mathcal{D}|} s_j^{(k)}, \quad s_j^{(k)} = \frac{v_j - w_j}{\max(v_j, w_j)},$$

   where $v_j$ and $w_j$ measure intra- and inter-cluster distances.

3. **Stratified Sampling with Maximal Marginal Relevance (MMR).** From each cluster, a subset is selected to balance topical relevance and diversity. For a candidate sample $x$, selected set $R$, and cluster $C$ with centroid $q_C$, the MMR score is

$$\text{MMR}(x) = \lambda \cdot \text{sim}(x, q_C) - (1 - \lambda) \cdot \max_{x_r \in R} \text{sim}(x, x_r),$$

   where $\lambda$ is a hyperparameter, and the next sample is chosen as $x^* = \arg\max_{x \in C \setminus R} \text{MMR}(x)$.

For binary datasets, we retain only one instance per question, either the positive or the negative, to avoid redundancy from semantically similar content. Further details on the final dataset composition and implementations are provided in Appendix Section B.1.

### 3.2.3 Rubric Generation

Many datasets lack explicit evaluation rubrics, which are essential to our framework for generating structured supervision. To address this, we automatically generate rubrics based on task type at inference time. Formally, for each sample $x^{(j)}$ in $\mathcal{D}_{20k}$, we transform the optional rubric $r_{opt}^{(j)}$ into a required rubric $r^{(j)}$, so the dataset becomes $\mathcal{D}_{20k} = \{(t^{(j)}, i^{(j)}, a^{(j)}, r^{(j)})\}_{j=1}^{20000}$. The rubrics are generated based on task type using the following method:

**Pair-wise and Binary Tasks.** We use templated prompts to generate rubric variations tailored to each format. To encourage generalization and mitigate overfitting, we randomize the rubric phrasing across three prompt variants. Full templates are listed in Appendices C.3 and C.4.

**Point-wise Tasks.** When original rubrics $r_{opt}^{(j)}$ are available (e.g., in FeedbackCollection), we reuse them. Otherwise, we generate task-specific rubrics targeting relevant evaluation criteria (e.g., helpfulness in Ultra-Feedback) using a few-shot prompting strategy with GPT-4.1 mini. Details on rubric prompting are available in Appendix C.1.

### 3.2.4 Explanation Trace Generation

Given the curated dataset $\mathcal{D}_{20k} = \{(t^{(j)}, i^{(j)}, a^{(j)}, r^{(j)})\}_{j=1}^{20,000}$, we distill natural language explanations using a reasoning distillation model. Specifically, we define a function:

$$\texttt{ReasoningModel}_\theta : (t^{(j)}, i^{(j)}, a^{(j)}, r^{(j)}) \longrightarrow \left(\text{reasoning\_trace}^{(j)}, \hat{s}^{(j)}, \hat{e}^{(j)}\right), \tag{5}$$

where $\texttt{ReasoningModel}_\theta$ is instantiated with DeepSeek-R1 (Guo et al., 2025). This model generates a natural language explanation (reasoning\_trace$^{(j)}$) along with short response of its predicted score $\hat{s}^{(j)}$ and a short justification span $\hat{e}^{(j)}$, following methodologies from prior work on explanation-based distillation (Shridhar et al., 2023; Vu et al., 2024). Prompting templates are provided in Appendix C. We define the final target for each sample $x^{(j)}$ as:

$$y^{(j)} = \text{reasoning\_trace}^{(j)} \oplus (\hat{s}^{(j)}, \hat{e}^{(j)}), \tag{6}$$

where $\oplus$ is string concatenation. Therefore, we define the dataset $\mathcal{D}_{20k}$ as $\mathcal{D}_{20k} = \{(x^{(j)}, y^{(j)})\}_{j=1}^{20000}$.

We find that approximately 20% of the reasoning traces are either overly verbose or contain repetitive content. For any example where $y^{(j)}$ exceeds 4,096 tokens, we summarize the reasoning trace using GPT-4.1 mini. The summarization preserves the core explanation while removing redundant content and maintains stylistic coherence with the original output. Details and heuristics for this step are provided in Appendix E.

As both the reasoning traces and their summaries are machine-generated, to verify the quality of the generated data, we conduct a human evaluation in Section F, where we assess the factual correctness and logical coherence of the original reasoning traces, as well as the faithfulness and style consistency of the trace summarizations.

### 3.2.5 Quality Filtering

Finally, to improve the quality of our training dataset while preserving the diversity of responses, we apply a two-stage filtering pipeline to the annotated dataset $\mathcal{D}_{20k}$.

**Step 1: Incorrect Prediction Filtering.** We discard examples for which the predicted score differs from the ground truth. Formally, we construct a filtered dataset $\mathcal{D}_{14k} \subset \mathcal{D}_{20k}$ such that for each retained example $(x^{(j)}, y^{(j)}) \in \mathcal{D}_{14k}$, we have $\hat{s}^{(j)} = s^{(j)}$, where $s^{(j)}$ is the true score for sample $x_{(j)}$. This ensures that all reasoning signals used for training are consistent with the gold labels. After this step, approximately 14,000 examples remain.

**Step 2: Triviality Filtering via Small Model Agreement.** To remove overly easy examples that provide a limited training signal, we evaluate each example in $\mathcal{D}_{14k}$ using our smallest model, Qwen3-4B (Yang et al., 2025). For each example $x^{(j)}$, we compute predictions across five decoding runs without chain-of-thought reasoning as $\{\hat{s}_{[1]}^{(j)}, \ldots, \hat{s}_{[5]}^{(j)}\} = \texttt{Qwen3-4B}(x^{(j)})$. If $\hat{s}_{[k]}^{(j)} = s^{(j)}$ for all $k \in \{1, \ldots, 5\}$, then we discard $x^{(j)}$ This results in the final dataset $\mathcal{D}_{4k} \subset \mathcal{D}_{14k}$, containing approximately 4,000 challenging and diverse training examples. We fine-tune our R3 models with both $\mathcal{D}_{4k}$ (-4K) and $\mathcal{D}_{14k}$ (-14K) to assess the impact of data size. Detailed dataset statistics are provided in Appendix Section B.1.

### 3.3 Reward Models Training and Evaluation

Given our generated training data, we further use supervised fine-tuning (SFT) to enhance the base model's reasoning capability as a reward model by minimizing the negative log-likelihood of reference responses. Given our training dataset $\mathcal{D} = \{(x^{(i)}, y^{(i)})\}_{i=1}^{N}$, where $x^{(i)}$ is prompt input previously introduced in eq. (1) and $y^{(i)} = (y_1^{(i)}, \ldots, y_{T_i}^{(i)})$ introduced in eq. (6) is the corresponding target sequence, the objective is the cross-entropy loss:

$$\mathcal{L}_{\text{SFT}}(\theta) = -\frac{1}{N} \sum_{i=1}^{N} \sum_{t=1}^{T_i} \log \pi_\theta\left(y_t^{(i)} \mid y_{<t}^{(i)}, x^{(i)}\right), \tag{7}$$

where $\pi_\theta(y_t \mid y_{<t}, x)$ denotes the model's conditional probability of token $y_t$ given the history $y_{<t}$ and prompt $x$, parameterized by $\theta$. By directly maximizing the log-likelihood of the ground-truth tokens, this loss encourages the base model to produce high-quality reasoning traces and the desired format for pair-wise comparisons or single-answer rewards. We further investigate lightweight fine-tuning via Low-rank Adaptation (LoRA) (Hu et al., 2022) on R3-4K data as part of our additional study to reduce training costs and data requirements while maintaining competitive performance.

Given that our tasks are within multiple domains, including but not limited to verifiable tasks like math and coding, reinforcement learning (RL) techniques are not directly applicable here. Nevertheless, it is possible to explore the effectiveness of using our trained model as a reward signal for enhancing other models' capabilities via RL, and we leave it as future work. For our R3 models, we mainly perform SFT on the Qwen3 model family (Yang et al., 2025) on the 4B, 8B, and 14B scales, along with Phi-4-reasoning-plus (Abdin et al., 2025). For fair comparison with other baselines, we also perform SFT on Qwen2.5-Instruct-7B (Team, 2024), DeepSeek-Distilled-Qwen-14B (Guo et al., 2025). For evaluation, we compare our open-source models against both open-source and proprietary baselines. Among open-source baselines, we primarily evaluate against `Prometheus-7B-v2.0` (Kim et al., 2024b), a popular rubric-based LLM-as-a-judge, and RM-R1 (Chen et al., 2025b) model suites, the most recent reasoning reward model work at the time of writing. For proprietary comparisons, we include DeepSeek-R1 (Guo et al., 2025), GPT-4.1 mini, GPT-o4 mini, and GPT-5 mini, strong closed models recognized for their general and reasoning-specific capabilities. Additionally, we include models that have reported results in their respective publications.

Finally, we evaluate the reward models across a diverse suite of benchmarks spanning multiple evaluation paradigms. For pairwise preference scoring, we use RM-Bench (Liu et al., 2024b) and RewardBench (Lambert et al., 2024b). For pointwise scoring, we employ XSUM (Narayan et al., 2018) and FeedbackBench (Kim et al., 2023). For binary classification tasks, we adopt MMLU-STEM (Hendrycks et al., 2021) and BBH (Suzgun et al., 2022). More details about the evaluation datasets and how we constructed them for evaluation can be found in Appendix Section B.2.

### 3.4 Policy Model Alignment Training and Evaluation

For policy model alignment, we use LLaMA 3.2–3B Instruct as the base policy model and perform Direct Preference Optimization (DPO) (Rafailov et al., 2023) on the HelpSteer3-Preference dataset (Wang et al., 2025a). We focus on DPO as RM-R1 is only applicable to preference data. In addition, to reduce confounding factors, we restrict training to the English portion of the dataset and to single-turn interactions, which ensures consistency with our evaluation setup.

We evaluate the aligned policy models using two widely adopted benchmarks: MT-Bench (Zheng et al., 2023) and WildBench (Lin et al., 2024), following the evaluation protocol of Wang et al. (2025b). We do not use AlpacaEval2 (Dubois et al., 2024), since WildBench contains more challenging and realistic prompts compared to AlpacaEval2 (Wang et al., 2025b). In both cases, model outputs are judged by GPT-4.1-mini, which we select over GPT-4o-Turbo due to its stronger empirical reliability as an evaluator and being cheaper. This setup allows us to measure the effectiveness of preference-based alignment in improving model helpfulness, robustness, and general instruction-following capability.

## 4 Results and Analysis

In this section, we present the overall performance of the reward modeling results. We mainly show results of R3 models trained with 14K samples. More detailed results, including results of our models that are trained using different strategies and results of other baseline models, can be found in the Appendix Section J.

### 4.1 Reward Models Evaluation

#### 4.1.1 Overall Performance

Table 2 and Table 3 highlight the strong performance of our R3 models on RM-Bench and RewardBench, showcasing the effectiveness of R3 models in pair-wise preference scoring under a training budget. Particularly

Table 2: Comparison of existing models with R3 trained with 14K samples on RM-Bench. **Bolded numbers** indicate the best-performing results between R3 models and baseline models. Proprietary models are bolded and compared independently.

| Model | Domain | | | | Difficulty | | | Overall |
|---|---|---|---|---|---|---|---|---|
| | Chat | Math | Code | Safety | Easy | Medium | Hard | Avg. |
| Prometheus-7B-v2.0 | 46.0 | 52.6 | 47.6 | 73.9 | 68.8 | 54.9 | 41.3 | 55.0 |
| JudgeLRM | 59.9 | 59.9 | 51.9 | 87.3 | 73.2 | 76.6 | 54.8 | 64.7 |
| RM-R1-Qwen-Instruct-7B | 66.6 | 67.0 | 54.6 | 92.6 | 79.2 | 71.7 | 59.7 | 70.2 |
| RM-R1-DeepSeek-Distilled-Qwen-7B | 64.0 | 83.9 | 56.2 | 85.3 | 75.9 | 73.1 | 68.1 | 72.4 |
| RM-R1-Qwen-Instruct-14B | **75.6** | 75.4 | 60.6 | 93.6 | 82.6 | 77.5 | 68.8 | 76.1 |
| RM-R1-Qwen-Instruct-32B | 75.3 | 80.2 | 66.8 | 93.9 | 86.3 | 80.5 | 70.4 | 79.1 |
| RM-R1-DeepSeek-Distilled-Qwen-14B | 71.8 | 90.5 | 69.5 | 94.1 | 86.2 | 83.6 | 74.4 | 81.5 |
| RM-R1-DeepSeek-Distilled-Qwen-32B | 74.2 | **91.8** | **74.1** | **95.4** | **89.5** | **85.4** | **76.7** | **83.9** |
| Qwen2.5-7B | 59.9 | 52.6 | 50.9 | 72.4 | 66.4 | 60.3 | 50.1 | 58.9 |
| DeepSeek-Distilled-Qwen-14B | 65.0 | 85.4 | 69.0 | 85.1 | 84.3 | 78.7 | 65.4 | 76.1 |
| Qwen3-4B | 64.3 | 84.0 | 62.0 | 85.6 | 82.6 | 74.5 | 64.8 | 74.0 |
| Qwen3-8B | 63.2 | 80.5 | 61.0 | 84.8 | 80.1 | 73.4 | 63.5 | 72.4 |
| R3 Models (Ours) | | | | | | | | |
| R3-Qwen2.5-7B | 66.8 | 82.0 | 65.0 | 87.0 | 83.8 | 76.8 | 64.9 | 75.2 |
| R3-DeepSeek-Distilled-Qwen-14B | 71.7 | 93.0 | 78.4 | 86.4 | 89.3 | 84.7 | 73.1 | 82.4 |
| R3-Qwen3-4B | 67.9 | 93.0 | 74.7 | 86.9 | 88.8 | 81.9 | 71.2 | 80.6 |
| R3-Qwen3-8B | 69.1 | 93.2 | 75.9 | 87.6 | 89.0 | 83.4 | 71.9 | 81.4 |
| R3-Qwen3-14B | 73.4 | **93.8** | **79.1** | **89.5** | **90.3** | **86.6** | **74.9** | **84.0** |
| R3-Phi-4-R$^+$-14B | **74.5** | 93.0 | 77.5 | 84.8 | 89.3 | 84.7 | 73.3 | 82.5 |
| Proprietary Models | | | | | | | | |
| GPT-4.1 mini | 67.6 | 73.0 | 71.3 | 90.7 | 87.0 | 78.4 | 61.7 | 75.7 |
| GPT-o4 mini | 77.6 | **93.0** | 80.8 | **93.4** | **92.0** | **88.7** | **78.0** | **86.2** |
| GPT-5 mini | **88.0** | 92.9 | **91.1** | 78.0 | 77.4 | 85.8 | 96.4 | 92.4 |
| DeepSeek-R1 | **78.6** | 66.2 | **81.9** | 88.7 | 86.9 | 82.2 | 67.3 | 78.8 |

in RM-bench, our models deliver remarkable results where even our smallest model, R3-Qwen3-4B, outperforms nearly all other reasoning models from RM-R1, with the exception of RM-R1-DeepSeek-Distilled-Qwen-14B and RM-R1-DeepSeek-Distilled-Qwen-32B. It also surpasses Prometheus-7B-v2.0, GPT-4.1 mini, and even DeepSeek-R1 as well, demonstrating its competitiveness. When comparing the same base models with RM-R1, our models R3-Qwen2.5-7B and R3-DeepSeek-Distilled-Qwen-14B outperform their RM-R1 counterparts up to 6.3 points.

Other variants of our R3 models also show competitive performance. For instance, R3-Qwen3-14B-LoRA-4k model (shown in the Appendix on Table 12) achieves the state-of-the-art performance, despite being trained with LoRA and smaller data. Between our R3-Qwen3-14B and R3-Phi-4-R$^+$ models, R3-Qwen3-14B models consistently outperforms R3-Phi-4-$R^+$ models in all aspects. Overall, the exceptional performance of our R3 models highlights the robustness and effectiveness of our training approach and meticulous data curation strategies.

In RewardBench, we achieve similarly impressive results. Our R3-Qwen3-4B models, despite being half the size of the RM-R1 7B models, outperform all RM-R1 7B models as well as Prometheus-7B-v2.0 by at least 1.8 points. Furthermore, the R3-Qwen3-4B-14k model surpasses GPT-4.1 mini by 0.5 points. When assessing our R3-Qwen3-14B models against the RM-R1 14B model families, the R3-Qwen3-14B-LoRA-4k model (shown in the Appendix on Table 13) surpasses RM-R1-DeepSeek-Distilled-Qwen-14B by 0.4 points, while matching the average performance of DeepSeek-R1. Between our R3-Qwen3-14B and R3-Phi-4-R$^+$ models, the R3-Qwen3-14B models generally outperform the R3-Phi-4-R$^+$ models, except in the Chat and Safety categories. Notably, our models demonstrate competitive performance even compared with DeepSeek-R1 under training budget constraints in terms of data and memory.

Table 3: Comparison of existing models with R3 on RewardBench using pair-wise scoring. **Bolded numbers** indicate the best-performing results between R3 models and baseline models. Proprietary models are bolded and compared independently.

| Models | Chat | Chat Hard | Safety | Reasoning | Avg. |
|---|---|---|---|---|---|
| Prometheus-7B-v2.0 | 90.2 | 45.6 | 75.8 | 74.6 | 71.6 |
| m-Prometheus-14B | 93.6 | 59.0 | 85.1 | 84.8 | 80.6 |
| JudgeLRM | 92.9 | 56.4 | 78.2 | 73.6 | 75.2 |
| SynRM | 38.0 | 82.5 | 74.1 | 87.1 | 70.4 |
| RM-R1-DeepSeek-Distilled-Qwen-7B | 88.9 | 66.2 | 78.4 | 87.0 | 80.1 |
| RM-R1-Qwen-Instruct-7B | **94.1** | 74.6 | 85.2 | 86.7 | 85.2 |
| RM-R1-Qwen-Instruct-14B | 93.6 | **80.5** | 86.9 | 92.0 | 88.2 |
| RM-R1-DeepSeek-Distilled-Qwen-14B | 91.3 | 79.4 | **89.3** | **95.5** | **88.9** |
| R3 Models (Ours) | | | | | |
| R3-Qwen2.5-7B | 91.4 | 73.8 | 85.1 | 90.6 | 85.2 |
| R3-DeepSeek-Distilled-Qwen-14B | 92.3 | 77.8 | 86.8 | 95.6 | 88.1 |
| R3-Qwen3-4B | 92.4 | 76.0 | 85.8 | 95.7 | 87.5 |
| R3-Qwen3-8B | 93.8 | 78.6 | 86.3 | 96.7 | 88.8 |
| R3-Qwen3-14B | 93.3 | **79.7** | **88.4** | **96.9** | **89.6** |
| R3-Phi-4-R$^+$-14B | **94.5** | 78.0 | 86.6 | 96.5 | 88.9 |
| Proprietary Models | | | | | |
| GPT-4.1 mini | **96.1** | 75.2 | 87.0 | 89.6 | 87.0 |
| GPT-o4 mini | 95.3 | **81.8** | 91.6 | **98.4** | **91.8** |
| GPT-5 mini | 95.3 | 81.6 | **92.0** | **98.4** | **91.8** |
| DeepSeek-R1 | 93.6 | 79.2 | 86.9 | 97.4 | 89.3 |

Table 4 presents the performance of our R3 models on point-wise assessment tasks, which are XSUM and FeedbackBench, along with binary tasks from BBH and MMLU-STEM. For XSUM, all R3 models consistently outperform DeepSeek-R1 and Prometheus-7B-v2.0 in terms of faithfulness, highlighting the effectiveness of binary assessment for R3 models. In terms of coherence and relevance, our R3-Phi-4-R$^+$ models perform the best among all open-source and proprietary models. On the generic rubric-based assessment from FeedbackBench, R3 models perform competitively against all baselines. While Prometheus-7B-v2.0 achieves the highest score on FeedbackBench, its relatively weaker performance on other benchmarks suggests that it may be better aligned with the specific rubric or distribution of FeedbackBench, rather than generalizing across diverse tasks.

For binary classification tasks such as BBH and MMLU-STEM, we observe that both larger model size and greater fine-tuning data improve performance, reflecting stronger reasoning capabilities. All of our R3 models outperform Prometheus-7B-v2.0, while R3-Qwen3-14B models surpass GPT-4.1 mini's performance. Overall, these results highlight the competitive and robust performance of R3 models across a range of point-wise and binary evaluation tasks.

### 4.1.2 Model Scaling and Efficiency

Figure 3 presents averaged results under three evaluation settings: binary, pairwise, and pointwise. Larger models generally demonstrate stronger reasoning and reward generation capabilities, and our R3 models show consistent improvements as parameter size increases, with some benchmarks exhibiting substantial gains. Within the same parameter size, R3 models achieve the best performance. Moreover, smaller variants of our R3 models can sometimes outperform larger models. For example, R3-Qwen3-4B outperforms Rise-judge-qwen2.5-32B and Prometheus-7B-v2.0 in the binary setting, while R3-Qwen3-14B outperforms RM-R1 32B in the pairwise setting. These results suggest that both our methodology and dataset are highly effective for training reward models in resource-constrained settings.

Table 4: Comparison of existing models with R3 on XSUM, FeedbackBench, BBH Binary, and MMLU-STEM Binary. **Bolded numbers** indicate the best-performing results between R3 models and baseline models. Proprietary models are bolded and compared independently.

| Models | XSUM | | | FeedbackBench | BBH Binary | MMLU-STEM Binary |
| | Acc. | Kendall Tau | | Kendall Tau | Acc. | Acc. |
| | Faithfulness | Coherence | Relevance | | | |
|---|---|---|---|---|---|---|
| Prometheus-7B-v2.0 | 60.7 | 0.12 | 0.16 | **0.79** | 54.0 | 56.5 |
| Selene-1-Mini-Llama-3.1-8B | 56.4 | 0.16 | 0.36 | 0.78 | 58.2 | 65.2 |
| RISE-Judge-Qwen2.5-7B | 66.4 | 0.29 | 0.32 | 0.68 | 63.1 | 76.9 |
| RISE-Judge-Qwen2.5-32B | 71.0 | **0.30** | **0.39** | 0.74 | **82.8** | 89.4 |
| R3 Models (Ours) | | | | | | |
| R3-Qwen2.5-7B | 67.5 | 0.33 | 0.35 | 0.70 | 81.1 | 88.3 |
| R3-DeepSeek-Distilled-Qwen-14B | 64.3 | **0.41** | 0.34 | 0.72 | 91.1 | 93.0 |
| R3-Qwen3-4B | 66.7 | 0.26 | 0.31 | 0.68 | 89.3 | 92.0 |
| R3-Qwen3-8B | 65.8 | 0.37 | 0.33 | 0.71 | 90.7 | 93.6 |
| R3-Qwen3-14B | **68.5** | 0.33 | 0.37 | 0.72 | 92.1 | **94.8** |
| R3-Phi-4-R$^+$-14B | 67.3 | 0.36 | 0.34 | 0.71 | **92.2** | 94.4 |
| Proprietary Models | | | | | | |
| GPT-4.1 mini | **72.6** | 0.07 | **0.38** | 0.69 | 91.0 | 93.3 |
| GPT-o4 mini | 69.1 | 0.16 | 0.30 | 0.65 | 93.2 | 95.3 |
| GPT-5 mini | 68.7 | **0.42** | **0.39** | 0.62 | **95.0** | **96.5** |
| DeepSeek-R1 | 60.4 | 0.35 | 0.38 | **0.72** | 94.0 | 96.2 |

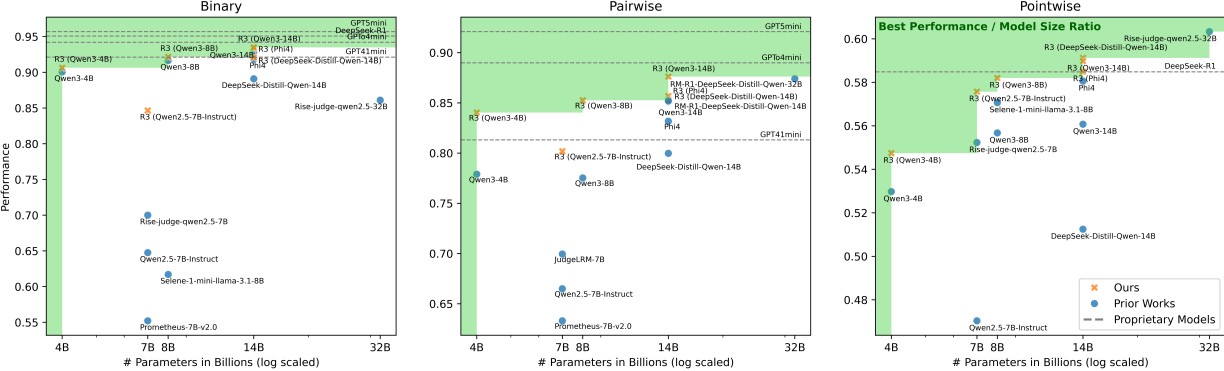

Figure 3: R3 models outperforms competitor models across differences model sizes in all data types.

### 4.1.3 Robustness

Among proprietary models, GPT-4o-mini outperforms DeepSeek-R1 on reward benchmarks involving pair-wise scoring, while DeepSeek-R1 demonstrates stronger performance on tasks such as XSUM, FeedbackBench, BBH, and MMLU-STEM. For open-weight models, our R3 models consistently outperform existing reward models, such as Prometheus-7B-v2.0 and all RM-R1 variants, across most benchmarks. The only exception is FeedbackBench, where Prometheus-7B-v2.0 performs exceptionally well. However, this suggests that Prometheus-7B-v2.0 is highly specialized rather than robust across tasks. In contrast, RM-R1 is more robust than Prometheus-7B-v2.0 but lacks flexibility in supporting diverse evaluation formats such as point-wise and binary scoring; Prometheus, meanwhile, supports only point-wise and pair-wise formats. Our R3 models offer both robustness and versatility, making it more suitable for general-purpose reward modeling.

### 4.1.4 Ablation Studies

We conduct an ablation study to assess the effectiveness of our overall dataset construction on different sampling strategies, dataset types, and supervision signals, with results summarized in Table 5. For efficiency, we apply LoRA (Hu et al., 2022) in all experiments using R3-Qwen3-14B. Our results indicate that random sampling consistently underperforms compared to diversity sampling. Among dataset types, pairwise supervision achieves the best results (82.1% on RM-Bench, 94.4% on MMLU-STEM), surpassing pointwise

Table 5: Ablation studies on dataset construction, employing the R3-QWEN3-14B model trained on a 14k-sample dataset using LoRA. **Bolded numbers indicate the best-performing results.**

|  | RM-Bench Overall Acc. | RewardBench Overall Acc. | BBH Overall Acc. | MMLU-STEM Overall Acc. |
|---|---|---|---|---|
| Random Sampling | 77.0 | 86.6 | 89.7 | 93.0 |
| Dataset |  |  |  |  |
|    Only Pairwise | **82.1** | **90.2** | **91.5** | **94.4** |
|    Only Pointwise | 80.0 | 86.0 | 90.1 | 93.4 |
|    Only Binary | 81.6 | 88.8 | 91.0 | 94.0 |
| No Rubric | 76.3 | 87.9 | 85.1 | 91.9 |
| No Explanation | **83.1** | **90.2** | **91.7** | **94.5** |
| No Reasoning | 71.2 | 82.6 | 79.8 | 88.2 |
| R3 | **83.5** | **90.2** | **91.9** | **94.5** |

Table 6: R3 Preference Optimization results using GPT-4.1-mini as the judge model. Our R3 models outperform a larger reward model, Llama-3.3-Nemotron-Super-49B-GenRM-Multilingual, despite the latter having over three times more parameters. **Bolded numbers** indicate the best performing results, while underlined numbers indicate the second-best performing results.

| Method | RM #Params | MT-Bench Overall | WildBench Overall | Creative Plan. | Data Analy. | Info. | Seek. | Coding |
|---|---|---|---|---|---|---|---|---|
| Llama 3.2 3B Instruct (Init Policy) |  | 5.75 | 15.70 | 43.10 | 25.12 | 3.65 | 34.44 | -5.97 |
| + DPO w/ RM-R1-DeepSeek-Distilled-Qwen-14B | 14B | 5.98 | 20.42 | 50.28 | 29.39 | 8.41 | 41.29 | -3.41 |
| + DPO w/ RM-R1-DeepSeek-Distilled-Qwen-32B | 32B | 6.01 | 22.28 | **54.32** | 31.66 | 9.37 | 43.91 | -2.56 |
| + DPO w/ Llama-3.3-Nemotron-Super-GenRM | 49B | **6.38** | 22.83 | 54.06 | 31.33 | 10.12 | **44.91** | -1.24 |
| + DPO w/ Llama-3.3-Nemotron-Super-GenRM-Multilingual | 49B | 6.20 | 23.30 | 52.71 | **31.93** | 11.43 | 44.90 | -0.57 |
| R3 Models (Ours) |  |  |  |  |  |  |  |  |
| + DPO w/ R3-QWEN3-4B-14K | 4B | 5.95 | 22.00 | 51.83 | 30.78 | 9.08 | 43.42 | -1.23 |
| + DPO w/ R3-QWEN3-8B-14K | 8B | 6.13 | 21.67 | 52.05 | 29.94 | 9.44 | 41.49 | -0.85 |
| + DPO w/ R3-QWEN3-14B-LoRA-4K | 14B | 6.20 | **23.45** | 52.57 | 31.62 | **12.03** | 42.98 | **1.04** |

and binary-only settings and improving relevance on XSUM. Supervision signals also have distinct effects: removing the rubric lowers BBH accuracy, excluding explanations reduces coherence, and eliminating reasoning traces causes the largest performance drop (e.g., 71.2% RM-Bench, 79.8% BBH), underscoring the importance of reasoning data. The full model (R3) achieves the best overall balance (83.5% RM-Bench, 94.5% MMLU-STEM, strong scores on coherence, relevance, and FeedbackBench). Although excluding explanations has a limited impact on accuracy, we retain them in R3 to enable more interpretable outputs.

## 4.2 Aligned Model Evaluation Results

Table 6 summarizes the performance of aligned models based on different reward models. Our R3 models consistently outperform reward models of similar size, with R3-QWEN3-14B-LoRA-4K achieving the best overall performance on WildBench, even surpassing the much larger Llama-3.3-Nemotron-Super-GenRM (49B) despite having only one-third of the parameters. On MT-Bench, R3 remains highly competitive, with only a small gap relative to Nemotron. This difference may be partly explained by Nemotron's training on HelpSteer3, a high-quality dataset featuring multi-turn interactions and multilingual coverage, unlike our R3-Dataset.

Compared to RM-R1 baselines, our models also deliver stronger results, especially on WildBench. Notably, even R3-QWEN3-4B outperforms the 14B RM-R1 model on WildBench while remaining competitive on MT-Bench, showcasing the efficiency of our approach. These findings demonstrate that R3 models consistently provide better performance within their parameter class while scaling effectively to close the gap with much larger models.

## 5   Related Work

**Rubric-Based Evaluation Models.**   Recent works leverage explicit rubrics to guide LLM evaluation. Kim et al. (2023) created FeedbackCollection, a fine-grained text evaluation finetuning dataset using detailed rubric for point-wise (direct accessment) evaluation. Kim et al. (2024b) followed-up by adding pair-wise evaluation to the training and found that weight merging performs better than training a jointly trained model. Likewise, LLM-Rubric (Hashemi et al., 2024) prompts an LLM on a human-authored multi-question rubric (e.g., dimensions like naturalness, conciseness, citation quality) and calibrates its outputs via a small model to match human judges. These rubric-driven methods yield fine-grained, interpretable assessments, but their reliance on laboriously constructed rubrics and reference solutions limits scalability and generality (Hashemi et al., 2024; Kim et al., 2023). By contrast, R3 eliminates the need for external rubrics, learning reward signals directly in a transparent form to enable broad, rubric-agnostic evaluation.

**Preference-Based Reward Models.**   Reward models learned from (implicit or explicit) human preferences—typically via RLHF or related methods—have become a standard alignment approach (Rafailov et al., 2023). In practice, however, learned RMs often exploit trivial cues: for instance, they tend to favor longer or more elaborate outputs (a well-known length bias) over brevity (Shen et al., 2023), and recent analyses show LLM evaluators even "self-recognize" and prefer their own generations over others of equal quality (Panickssery et al., 2024). Zhu et al. (2025) further demonstrate "model preference bias" in RMs, whereby certain models' outputs are systematically overvalued. Such biases and spurious correlations undermine fairness and generalization. New techniques mitigate these issues: DPO recasts RLHF in a simpler optimization framework (Rafailov et al., 2023), and Vu et al. (2024) train FLAMe on 5M+ human judgements across 100+ tasks, achieving stronger OOD generalization and even outperforming GPT-4 on reward-modeling benchmarks. Despite these advances, preference-trained RMs remain large, opaque models tied to specific data, motivating R3's interpretable, rubric-free reward formulation as a more transparent alternative.

**LLM-as-a-judge Framework.**   Using a pretrained LLM itself as the evaluator has gained popularity due to its zero-shot flexibility (Kim et al., 2024b). However, numerous studies reveal reliability issues. For instance, Wang et al. (2024b) found that simply altering the order of candidate responses can drastically flip an LLM judge's ranking, making one model appear vastly superior or inferior. More broadly, LLM evaluators suffer from hallucinations and entrenched biases; e.g., Panickssery et al. (2024) show LLM judges systematically score their own outputs higher than others' ("self-preference" bias), and Zhu et al. (2025) observe strong model-specific scoring bias in LLM-based evaluation. These flaws undermine trust and consistency in LLM-as-judge systems. R3 addresses these gaps by providing a fully interpretable reward model that avoids opaque LLM judgments and fixed rubrics, while supporting a broader range of evaluation types for greater flexibility.

## 6   Conclusion

In this paper, we introduce R3, a novel reward modeling framework that is rubric-agnostic, generalizable across evaluation dimensions, and capable of producing interpretable, reasoning-based score assignments. Leveraging reasoning distillation, targeted dataset curation, and a two-stage quality filtering pipeline, R3 addresses key limitations of prior reward models in terms of interpretability, controllability, and generalizability. Despite using training datasets that are an order of magnitude smaller than those of many baselines, R3 models matches or surpasses state-of-the-art performance in reward model benchmarks. Our experiments demonstrate the method's strong training efficiency and scalability, including effective use of compute-efficient techniques. Policy models that are aligned with our R3 models also perform better compared to those trained with baseline reward models. By enabling more transparent and adaptable evaluation, R3 advances robust alignment with diverse human values and real-world applications—paving the way for more trustworthy and versatile reward models.

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

## A Limitations

In our experiments, we limit our exploration to models with up to 14B parameters due to resource constraints. We also include smaller models in our study, aiming to shed light on scaling behavior and its impact on performance. Larger models, such as those with 32B parameters or more, are left for future investigation.

## B Details about Datasets

### B.1 Details About Reward Model Training Datasets

#### B.1.1 Details About Dataset Sampling

The following describes a more detailed process of the three-stage sampling process to extract the most diverse examples:

1. **Embedding and Preprocessing**. We begin by embedding each instance using a semantic representation that combines its task instruction and input text to capture the sample's semantics across topics. Specifically, we use the `gte-Qwen2-7B-instruct` model (Li et al., 2023) to compute embeddings over $h(x^{(j)}) = t^{(j)} \oplus i^{(j)}$, where $\oplus$ denotes string concatenation. The resulting embedding $Emb(h(x^{(j)})) = q^{(j)}$ is used to measure similarity and diversity in semantic space during clustering.

2. **Cluster Determination and Assignment**. To identify an appropriate number of groups $k^* \in \{k_{\min}, \ldots, k_{\max}\}$, we select the value of $k$ that maximizes the average Silhouette score (Shahapure & Nicholas, 2020). Here we choose $k_{min} = 3$ and $k_{max} = 10$. If the dataset includes labeled subcategories (e.g., topics or task types), clustering is applied independently within each subcategory to preserve intra-category diversity. The Silhouette score for a sample $x^{(j)}$ is defined as $s_j = \frac{v_j - w_j}{\max(v_j, w_j)}$ where $v_j$ is the mean distance between $x_j$ and other points in the same cluster, and $w_j$ is the mean distance to the nearest cluster not containing $x^{(j)}$. We select the optimal number of clusters $k^*$ by

$$k^* = \arg \max_{k \in \{k_{\min}, \ldots, k_{\max}\}} \frac{1}{|\mathcal{D}|} \sum_{j=1}^{|\mathcal{D}|} s_j^{(k)}, \tag{8}$$

   where $s_j^{(k)}$ is the Silhouette score of sample $x^{(j)}$ under the clustering configuration with $k$ clusters.

3. **Stratified Sampling with Maximal Marginal Relevance (MMR)**. We perform stratified sampling from each cluster with a minimum of 10 samples per cluster. For each cluster $C$ with centroid $q_C$:

   - We retain the first 25% of samples based on the closest to the cluster centroid, to ensure topical relevance, i.e., $R_{closest} = Top_{\lfloor 0.25 \cdot |C| \rfloor}\{x \in C \mid \|Emb(x) - q_C\|_2\}$;
   - The next 75% of the samples are selected via MMR, which balances relevance and diversity among the already selected samples. Let $R$ denote the set of already selected examples, in which initially $R = R_{closest}$. To sample the next candidate $x \in C \setminus R$, we compute the MMR score as:

$$\text{MMR}(x) = \lambda \cdot \text{sim}(x, q_C) - (1 - \lambda) \cdot \max_{x_r \in R} \text{sim}(x, x_r), \tag{9}$$

   where $\text{sim}(\cdot, \cdot)$ denotes cosine similarity, and $\lambda \in [0, 1]$ is a tunable trade-off parameter, in which we set $\lambda = 0.5$ to balance relevance and diversity. The next selected example is $x^* = \arg \max_{x \in C \setminus R} \text{MMR}(x)$.

#### B.1.2 Dataset Size and Composition

Table 7 showcases the composition of $\mathcal{D}_{20k}$, $\mathcal{D}_{14k}$, and $\mathcal{D}_{4k}$.

Table 7: Dataset size and composition of the top 7 source datasets at each stage of filtering. FC = Feedback Collection, PC = Preference Collection.

| | Count | Tulu3 | AceCodePair | Math-step-DPO | FC | PC | UltraFeedback | Skywork |
|---|---|---|---|---|---|---|---|---|
| $\mathcal{D}_{20k}$ | 20,000 | 0.18 | 0.15 | 0.15 | 0.13 | 0.10 | 0.10 | 0.10 |
| $\mathcal{D}_{14k}$ (Filter Step 1) | 13,772 | 0.19 | 0.20 | 0.21 | 0.09 | 0.07 | 0.06 | 0.11 |
| $\mathcal{D}_{4k}$ (Filter Step 2) | 3,949 | 0.13 | 0.28 | 0.19 | 0.12 | 0.03 | 0.12 | 0.05 |

Table 8: Length (white-space separated word count) distribution of our dataset. Response length includes DeepSeek-R1 thinking tokens along with the short response, which contains an explanation and the score assigned.

| | Prompt Length | Response Length |
|---|---|---|
| $\mathcal{D}_{20k}$ | $504 \pm 302$ | $850 \pm 847$ |
| $\mathcal{D}_{14k}$ (Filter Step 1) | $497 \pm 413$ | $729 \pm 538$ |
| $\mathcal{D}_{4k}$ (Filter Step 2) | $442 \pm 224$ | $851 \pm 599$ |

### B.1.3 Prompt and Response Length

In Table 8 we document the length distribution of our dataset.

### B.1.4 Label Distribution

In Table B.1.4 we show the label distribution of our dataset across different filtering stages. Our raw dataset has balanced distribution within each evaluation type. In $\mathcal{D}_{14k}$ (Filter Step 1), binary labels are slightly biased towards "false" and pair-wise labels are slightly biased towards "Response 1". In $\mathcal{D}_{4k}$ (Filter Step 2), binary labels are slightly biased towards "true" and and pair-wise labels are slightly biased towards "Response 1". Point-wise scores are also biased towards middle values (i.e., "3").

Table 9: Dataset label statistics distribution across the filtering process.

| | Binary | | Pair-wise | | Point-wise | | | | |
|---|---|---|---|---|---|---|---|---|---|
| | true | false | resp. 1 | resp. 2 | 1 | 2 | 3 | 4 | 5 |
| $\mathcal{D}_{20k}$ | 0.024 | 0.026 | 0.340 | 0.335 | 0.047 | 0.053 | 0.055 | 0.058 | 0.062 |
| $\mathcal{D}_{14k}$ (Filter Step 1) | 0.024 | 0.031 | 0.429 | 0.354 | 0.040 | 0.036 | 0.021 | 0.038 | 0.027 |
| $\mathcal{D}_{4k}$ (Filter Step 2) | 0.033 | 0.022 | 0.365 | 0.304 | 0.035 | 0.048 | 0.700 | 0.061 | 0.046 |

## B.2 Details About Reward Model Evaluation Datasets

RM-Bench (Liu et al., 2024b) contains 1.3K instances across four domains—Chat, Safety, Math, and Code—each with prompts at three difficulty levels. RewardBench (Lambert et al., 2024b) includes 3K preference pairs in four categories: Chat, Chat-Hard, Safety, and Reasoning, where we report both category-wise and overall accuracy. For summarization, we use a human-annotated subset of XSUM (Narayan et al., 2018) with labels on faithfulness (binary), coherence, and relevance (Likert 1–5) following Zhang et al. (2024b); we compute accuracy for faithfulness and Kendall-Tau (Sen, 1968) correlations for coherence and relevance. FeedbackBench (Kim et al., 2023), the test split of FeedbackCollection, contains 1K rubrics, 200 instructions, and 1K responses, evaluated via Kendall-Tau correlation. For STEM reasoning, we construct MMLU-STEM Binary (Hendrycks et al., 2021) from a subset.[1]

**RM-Bench**  (Liu et al., 2024b) is a reward model evaluation benchmark consisting of 1.3K instances that cover four domains: Chat, Safety, Math, and Code. Each instance consists of three prompts categorized by

---

[1]https://huggingface.co/datasets/TIGER-Lab/MMLU-STEM

difficulty level: easy, medium, and hard. We measure the accuracy on each domain and difficulty level, along with the overall average accuracy.

**RewardBench.** (Lambert et al., 2024b) is a popular reward model evaluation benchmark consists of 3K instances of preference pairs on four categories: Chat, Chat-Hard, Safety, Reasoning. We measure the accuracy on each category along with the overall average accuracy.

**XSUM.** (Narayan et al., 2018) is a news summarization dataset. For our evaluation, we use a subset that has been annotated by human evaluators across three criteria: faithfulness (binary), coherence (Likert scale 1–5), and relevance (Likert scale 1–5), following the annotation protocol of Zhang et al. (2024b). We measure the Kendall-Tau (Sen, 1968) correlation for coherence and relevance, while we measure accuracy for faithfulness.

**FeedbackBench.** (Kim et al., 2023) is the test split of FeedbackCollection introduced with the Prometheus model for evaluating point-wise tasks. It contains 1K score rubrics, 200 instructions, and 1K responses that do not overlap with the train data. We measure the Kendall-Tau (Sen, 1968) correlation as previously done by  Kim et al. (2023).

**MMLU-STEM Binary.** (Hendrycks et al., 2021) is a STEM-subject related subset[2] of the MMLU benchmark with multiple-choice questions from various branches of knowledge. Given four potential choices and one correct answer, we convert it to a binary evaluation task. For each original question, we evaluate model's response given the correct and separate a randomly selected incorrect answer. We measure the overall accuracy, along with the accuracy on each subject. Unless otherwise specified, all references to MMLU-STEM in this work refer to our MMLU-STEM Binary benchmark.

**BBH Binary.** (Suzgun et al., 2022) is a collection of 27 non-trivial reasoning-like tasks sourced from BigBench (bench authors, 2023) with a total of 6.7K instances. The format of the tasks can be multiple choice or short string completion. Similar to MMLU-STEM, we include a copy of the data with the correct response and a copy with the incorrect response. Details of the dataset generation process is in Appendix H. We measure the overall accuracy. Unless otherwise specified, all references to BBH in this work refer to our BBH Binary benchmark.

### B.3  Details About Policy Alignment Evaluation Datasets

MT Bench contains 80 samples from 8 diverse categories (Writing, Roleplay, Extraction, STEM, Humanities, Reasoning, Math and Coding), each with two turns. WildBench, on the other han,d contains 1024 diverse real-world variable-turn prompts relating to Creative, Planning/Reasoning, Data Analysis/Math, Information/Advice seeking and Coding/Debugging.

## C  Prompt Template

### C.1  Rubric Generation Template

For point-wise tasks, we generate rubric with Likert score from 1 to 5 using the following template.

> **Rubric generation template**
>
> You are an expert evaluator. Given a defined task, analyze the task and create a rubric using a Likert scale from 1 to 5 to that will help to perform the given task.
> Please follow these steps:
> 1. Explain the criteria for distinguishing between the scores (e.g., how a score of 1 differs from a score of 5).

---

[2]https://huggingface.co/datasets/TIGER-Lab/MMLU-STEM.

2. Based on your analysis, generate a rubric in JSON format with the Likert scale ranging from 1 to 5, including descriptions for each score.
3. Ensure that the rubric is clear, actionable, and covers key aspects of the task.

### TASK
{task_instruction}

### INPUT
{input/question}

### EXAMPLE RUBRICS (Unrelated Tasks)
{sample_rubrics}

### RUBRIC FOR CURRENT TASK

## C.2 Point-wise Evaluation

For point-wise tasks where the judge model needs to assign a score for a response from 1-5, we use the following template.

**Pointwise evaluation prompt template**

Evaluate the response based on the given task, input, response, and evaluation rubric.
Provide a fair and detailed assessment following the rubric.

### TASK
{task_instruction}

### INPUT
{input/question}

### RESPONSE
{response}

### EVALUATION RUBRIC
1: {score_of_1_description}
2: {score_of_2_description}
3: {score_of_3_description}
4: {score_of_4_description}
5: {score_of_5_description}

### OUTPUT FORMAT
Return a JSON response in the following format:

{
"explanation": "Explanation of why the response received a particular score",
"score": "Score assigned to the response based on the rubric between 1 to 5"
}

### EVALUATION

### C.3 Pair-wise Evaluation

For pair-wise tasks where the judge model needs to compare against two responses, we use the following template.

---

**Pairwise evaluation prompt template**

Evaluate the response based on the given task, input, two responses, and evaluation rubric.
Provide a fair and detailed assessment following the rubric.

### TASK
{task_instruction}

### INPUT
{input/question}

### RESPONSE 1
{response_1}

### RESPONSE 2
{response_2}

### EVALUATION RUBRIC
Response 1: Response 1 provided better response, rejecting Response 2.
Response 2: Response 2 provided better response, rejecting Response 1.

### OUTPUT FORMAT
Return a JSON response in the following format:

{
"explanation": "Explanation of why one response is preferred over the other",
"score": "Final selection between 'Response 1' or 'Response 2'"
}

### EVALUATION

---

For rubrics, we include three variations and uniformly randomly sample from them when creating our dataset.

---

**Pairwise evaluation rubric variation 1**

{
"response_1": "Response 1 is the preferred choice over Response 2.",
"response_2": "Response 2 is the preferred choice over Response 1."
}

---

**Pairwise evaluation rubric variation 2**

{
"response_1": "Response 1 provided better response, rejecting Response 2.",
"response_2": "Response 2 provided better response, rejecting Response 1."
}

---

---

**Pairwise evaluation rubric variation 3**

```
{
"response_1": "Response 1 is superior, meaning Response 2 is not chosen.",
"response_2": "Response 2 is superior, meaning Response 1 is not chosen."
}
```

---

### C.4 Binary Evaluation

For binary tasks where the judge model needs to classify true or false to the response, we use the following template.

---

**Binary evaluation prompt template**

Evaluate the response based on the given task, input, response, and evaluation rubric.
Provide a fair and detailed assessment following the rubric.

### TASK
{task_instruction}

### INPUT
{input/question}

### RESPONSE
{response}

### EVALUATION RUBRIC
true: The response accurately reflects the correct answer based on the input.
false: The response does not accurately reflect the correct answer based on the input.

### OUTPUT FORMAT
Return a JSON response in the following format:

```
{
"explanation": "Explanation of why the answer is true or false",
"score": "Final boolean answer between true or false"
}
```

### EVALUATION

---

For rubrics, we include three variations and uniformly randomly sample from them when creating our dataset.

---

**Binary evaluation rubric variation 1**

```
{
"true": "The answer correctly corresponds to the given input.",
"false": "The answer does not correctly correspond to the given input."
}
```

---

---

**Binary evaluation rubric variation 2**

```
{
"true": "The response accurately reflects the correct answer based on the input.",
"false": "The response does not accurately reflect the correct answer based on the input."
}
```

**Binary evaluation rubric variation 3**

```
{
"true": "The answer is appropriate and correctly derived from the input.",
"false": "The answer is inappropriate or incorrectly derived from the input."
}
```

## D  Example Prompts and Responses

### D.1  Point-wise Evaluation

**Prompt example for pointwise evaluation (sourced from FeedbackCollection)**

Evaluate the response based on the given task, input, response, and evaluation rubric.
Provide a fair and detailed assessment following the rubric.

### TASK
Does the final conclusion drawn by the response hold up to logical scrutiny and provide a correct solution for an instruction with a definite answer?

### INPUT
Imagine a situation where there is a debate going on regarding the total number of prime numbers between 1 and 100. Your task is to determine the correct count and also provide the list. The debate is currently revolving around three different answers - 25, 26 and 27 prime numbers.

### RESPONSE
The total number of prime numbers between 1 and 100 is definitely 30. Prime numbers are those numbers that only have 1 and the number itself as factors. We don't need to list down each of them because it's already known that there are 30 prime numbers between 1 and 100. It's simple mathematics, and there is no need for any debate on this matter. The discussion around the number being 25, 26, or 27 is irrelevant and baseless. So, the conclusion is there are 30 prime numbers between 1 and 100.

### EVALUATION RUBRIC
1: The concluding answer from the model is entirely false and devoid of logical foundation.
2: The concluding answer from the model has major flaws that seriously compromise its validity.
3: The concluding answer from the model bears considerable mistakes that demand significant rectification.
4: The concluding answer from the model has slight inaccuracies, but these are simple to fix and do not greatly affect its overall validity.
5: The concluding answer from the model is wholly correct and logically sound.

### OUTPUT FORMAT
Return a JSON response in the following format:

---

```
{
"explanation": "Explanation of why the response received a particular score",
"score": "Score assigned to the response based on the rubric between 1 to 5"
}

### EVALUATION
```

---

**Example response from R1 (summarized) used for SFT**

<think>
Okay, let me evaluate this response. The user asked about the number of primes between 1 and 100, with the debate being 25, 26, or 27. The response says there are 30 primes, which is definitely wrong. The actual count is 25. The response claims it's "simple mathematics" and dismisses the debate, but that's incorrect. The answer provided is entirely false, and the reasoning is flawed because they didn't list or verify the primes. The rubric says a score of 1 is for entirely false and no logical foundation. This fits because the answer is wrong and the logic is absent. So, score 1.
</think>
```
{
"explanation": "The response concludes there are 30 prime numbers between 1 and 100, which is
factually incorrect. The actual number is 25. The claim that 'it's already known there are 30 primes'
is baseless, and the dismissal of the debate over 25, 26, or 27 as 'irrelevant' ignores the correct answer
(25). The response fails to provide logical reasoning, a correct list, or acknowledgment of established
mathematical facts. This demonstrates a complete lack of validity and logical foundation.",
"score": "1"
}
```

## D.2 Pair-wise Evaluation

**Prompt example for pairwise evaluation (sourced from Tulu-3-sft)**

Evaluate the response based on the given task, input, two responses, and evaluation rubric.
Provide a fair and detailed assessment following the rubric.

### TASK
Evaluate the factual accuracy of the response. Consider whether the information provided is correct,
up-to-date, and free from errors or misconceptions.

### INPUT
Come up with a question and reasoning that would justify this answer: Raphael Sbarge Your ENTIRE
response should be in Chinese, no other language is allowed. Come up with a question and reasoning
that would justify this answer: Raphael Sbarge

### RESPONSE 1
Question: 《Blade》这部吸血鬼题材的电影中，扮演主角V的演员是谁？

Reasoning: 电影《Blade》是一部著名的吸血鬼题材作品，Raphael Sbarge在这部电影中饰演
了主角V，这是一个非常有影响力的角色。因此，这个问题和回答完美契合，Raphael Sbarge正是该
角色的扮演者。

### RESPONSE 2
问题：谁是演 'Once Upon a Time' 中的 "金Pinocchio/Archie Hopper" 一角的演员？

理由：Raphael Sbarge 是美国演员，他最著名的角色之一就是美国电视剧 'Once Upon a Time' 中的金Pinocchio/Archie Hopper。

### EVALUATION RUBRIC
Response 1: Response 1 provided better response, rejecting Response 2.
Response 2: Response 2 provided better response, rejecting Response 1.

### OUTPUT FORMAT
Return a JSON response in the following format:

{
"explanation": "Explanation of why one response is preferred over the other",
"score": "Final selection between 'Response 1' or 'Response 2'"
}

### EVALUATION

---

**Pairwise example response from R1 (summarized) used for SFT**

<think>
好的，我现在需要评估两个回答，看看哪个在事实准确性上更好。任务是根据提供的输入，判断哪个回答更准确、最新且没有错误。输入是让用户想出一个问题和理由来证明答案是Raphael Sbarge。

首先看Response 1的问题："《Blade》这部吸血鬼题材的电影中，扮演主角V的演员是谁？"理由里说Raphael Sbarge在这部电影中饰演了主角V。但根据我的了解，《Blade》系列电影的主角Blade是由Wesley Snipes扮演的，而"V"可能是指另一部电影《V字仇杀队》里的主角V，但那个是由Hugo Weaving饰演的。Raphael Sbarge确实参演过《Blade》吗？我需要确认一下。查资料发现，Raphael Sbarge确实在1998年的《刀锋战士》（Blade）中出演过一个角色，但并不是主角V，而是次要角色Dr. Curtis Webb。所以Response 1的问题和理由存在错误，把主角V的扮演者错误地归为Sbarge，这显然是错误的。

接下来看Response 2的问题："谁是演 'Once Upon a Time' 中的 '金Pinocchio/Archie Hopper' 一角的演员？"理由指出Sbarge在该剧中扮演这个角色。根据资料，Raphael Sbarge确实在《童话镇》（Once Upon a Time）中扮演了Archie Hopper，也就是木偶匹诺曹的人类形态。这个信息是正确的，所以Response 2的问题和理由都是事实正确的。

根据评分标准，应该选择事实准确的回答。显然Response 2正确，而Response 1有错误，因此Response 2更优。
</think>```json
{
"explanation": "Response 1存在事实错误：《Blade》系列电影的主角V并非由Raphael Sbarge饰演（实际由Wesley Snipes饰演），且该电影中Raphael Sbarge仅饰演次要角色Dr. Curtis Webb。Response 2则正确指出Raphael Sbarge在《Once Upon a Time》中饰演Archie Hopper/金Pinocchio，这一信息符合事实且无错误。因此，Response 2在事实准确性上明显优于Response 1。",
"score": "Response 2"
} ```

### D.3 Binary Evaluation

---

**Prompt example for binary evaluation (sourced from evouna)**

Evaluate the response based on the given task, input, response, and evaluation rubric.
Provide a fair and detailed assessment following the rubric.

### TASK
Your task is to determine whether the given answer response is correct based on the query input.

### INPUT
who was the killer in the movie i know what you did last summer

### RESPONSE
Ben Willis

### EVALUATION RUBRIC
true: The response accurately reflects the correct answer based on the input.
false: The response does not accurately reflect the correct answer based on the input.

### OUTPUT FORMAT
Return a JSON response in the following format:

{
"explanation": "Explanation of why the answer is true or false",
"score": "Final boolean answer between true or false"
}

### EVALUATION

---

**Prompt example for binary evaluation (sourced from evouna)**

<think>
Okay, let's tackle this evaluation. The user is asking whether the killer in the movie "I Know What You Did Last Summer" is Ben Willis, as stated in the response. First, I need to recall the plot of the movie to verify this. From what I remember, the movie revolves around a group of friends who hit someone with their car and then cover it up. The killer is later revealed to be seeking revenge for that incident. The main antagonist is indeed Ben Willis, who is also known as the Fisherman. He's the father of the person they hit, and he faked his own death to frame someone else. So, the response "Ben Willis" is correct. The answer should be true.
</think>
{
"explanation": "In the movie 'I Know What You Did Last Summer,' the killer is revealed to be Ben Willis, also known as the Fisherman. He is the father of the victim the group accidentally hit with their car and covers up. Ben Willis fakes his death and seeks revenge, making the response accurate.",
"score": "true"
}

---

## E    Explanation Trace Summarization Details

First, we perform inference using the model to obtain the initial reasoning trace. This training trace is then passed through the model once more; conditioned on the prompt shown in "Prompt for Summarization

Tracing" to generate a concise version. The second inference produces a shortened reasoning trace by removing redundant or unnecessary reasoning steps while preserving the original tone, style, and logical progression.

---

**Prompt for Reasoning Trace Summarization**

Shorten the following reasoning trace by removing redundant or unnecessary thinking loops while preserving the exact same tone, style, and progression of thought. Output only the shortened reasoning trace without any explanation.

{DeepSeek-R1 Reasoning Trace}

---

## F   Human Evaluation of Reasoning Traces

We recruit five annotators to annotate approximately 2% of $\mathcal{D}_{4k}$, which was stratified sampled from various dataset sources, to verify both the reliability of the reasoning traces and the quality of the trace summarization. Details of the annotations setup, metrics we use to annotate, the experiments, and results are in Appendix I. We find on average the reasoning traces score $2.9 \pm 0.2$ (out of 3, higher better) in factual correctness, $2.8 \pm 0.2$ in logical coherence (n=93). The faithfulness of the summary scores averages $2.8 \pm 0.5$ and the style consistency scores $2.7 \pm 0.4$ (n=84). These results confirm the high quality reasoning traces used in our dataset.

## G   Training Hyper-parameters

For all of our experiments, we use 4 A800 80GB GPUs.

We use LLaMA-Factory Zheng et al. (2024) to perform SFT for all R3 models. We set the maximum sequence length to 8192, with a learning rate of $1e-5$, trained for 5 epochs using a cosine learning rate scheduler. The batch size per device is 16. For R3 LoRA models, we use LoRA rank of 64 and alpha of 128. For inference, we use vLLM Kwon et al. (2023) using the recommended inference configuration from Qwen3 and Phi-4-reasoning-plus.

## H   Evaluation Prompt

Since RewardBench and FeedbackBench are of pair-wise and point-wise evaluation format, they do not require extra processing to format into our prompt template. For both MMLU-STEM and BBH, since we are converting them to binary evaluation, we need to sample negative responses to augment the dataset.

The original MMLU-STEM consists of multiple-choice questions. We simply randomly sample a wrong answer as the negative. For subtasks of BBH that are also in the format of multiple-choice questions, we do the same.

There are four tasks that require custom adaptation for negative label sampling:

**DyckLanguages**   is a task where models are tasked to complete un-closed parentheses of different types. To sample negatives, with equal chance, we randomly delete, swap, or insert a symbol that appears in the context.

**WordSorting**   is a task where models are tasked to sort a set of unordered words. We randomly swap a pair of words from the target order to create the negative.

**MultistepArithmeticTwo**   is a task where models are expected to perform arithmetic calculations involving 8 single-digit operands. We calculate the mean and standard deviation of the label distribution, and randomly sample a number within the distribution.

**ObjectCounting** is a task where models are expected to count the number of objects (possibly a subset of all mentioned objects) mentioned in a sentence. We calculate the mean and standard deviation of the label count distribution, and randomly sample a number within the distribution.

## I Human Annotation Details

We stratified-sample 100 instances of data, and have the authors of the paper annotate the quality of the reasoning and reasoning summarizations. In total we have 5 annotators, annotating a total of around 2% of $\mathcal{D}_{4k}$.

### I.1 Reliability of Reasoning Trace

To ensure reasoning trace is reliable, we define two metrics **Factual Correctness** and **Logical Coherence** to ensure consistent labeling:

**Factual Correctness** (Scale: 1–3) assesses whether the statements in the reasoning trace are true and supported by external knowledge or evidence. When scoring, treat retrievable evidence or commonsense facts as acceptable grounding.

1. (Incorrect) Contains one or more clear factual errors or hallucinations that undermine the trace. May lead to incorrect conclusions or mislead the model.

2. (Partially Correct) Most statements are accurate, but minor factual errors or unverifiable claims exist. Does not change the final conclusion, but may reduce trace reliability.

3. (Fully Correct) All statements are factually accurate and supported by known facts, context, or ground truth. No hallucinations or inaccuracies.

**Logical Coherence** measures whether the reasoning steps logically follow from each other and form a coherent argument or thought process. Judge based on internal consistency, not factuality. A trace can be factually wrong but still logically coherent.

1. (Incoherent) Trace is illogical, disjointed, or internally inconsistent. Steps may contradict, skip crucial logic, or appear arbitrary.

2. (Somewhat Coherent) Mostly logical, but has minor gaps, unclear transitions, or weak justifications. Still understandable, but less robust as supervision.

3. (Fully Coherent) All steps follow logically and consistently. No missing steps, contradictions, or unjustified jumps in reasoning. A smooth, interpretable chain.

In Table 10 we show detailed annotation results across annotators.

Table 10: Human annotation results on reasoning trace **factual correctness** and **logical coherence** (out of 3, higher better).

|  | Annotator 1 | Annotator 2 | Annotator 3 | Annotator 4 | Average |
|---|---|---|---|---|---|
| Factual Correctness | $3 \pm 0.2$ | $3 \pm 0$ | $2.9 \pm 0.3$ | $2.8 \pm 0.5$ | $2.9 \pm 0.2$ |
| Logical Coherence | $2.9 \pm 0.4$ | $2.6 \pm 0.7$ | $2.9 \pm 0.3$ | $2.7 \pm 0.5$ | $2.8 \pm 0.2$ |
| Count | 27 | 10 | 28 | 28 | - |

### I.2 Reasoning Trace Summary Quality

During dataset curation, we use GPT-4.1 mini to summarize the reasoning traces that are too long. We want to measure **faithfulness** and **style similarity**.

**Faithfulness**   measures how well the summary covers the ideas of the original reasoning trace

1. (Unfaithful) Omits key reasoning or introduces incorrect logic. Could mislead a model or change the original meaning.

2. (Partially Faithful) Minor omissions or slightly altered emphasis, but preserves the general logic and outcome. Acceptable for training.

3. (Fully Faithful) Captures all core and necessary reasoning steps accurately. No hallucinations, distortions, or omissions of crucial logic.

**Style Similarity**   includes similar tone, level of formality, structured markers ("first", "therefore"), or domain-specific phrasing.

1. (Completely different) Omits all tone, level of formality, etc. from original trace

2. (Somewhat similar style) Somewhat similar in terms of tone, level of formality, etc. from original trace

3. (Same style) Same style with the original reasoning trace

In Table 11 we show detailed annotation results across annotators.

Table 11: Human annotation results on reasoning trace summary **faithfulness** and **style similarity** (out of 3, higher better).

|  | Annotator 1 | Annotator 2 | Annotator 3 | Annotator 4 | Annotator 5 | Average |
|---|---|---|---|---|---|---|
| Faithfulness | $2.6 \pm 0.8$ | $2.8 \pm 0.5$ | $2.8 \pm 0.4$ | $2.7 \pm 0.8$ | $3.0 \pm 0.0$ | $2.8 \pm 0.5$ |
| Style similarity | $2.5 \pm 0.6$ | $3.0 \pm 0.2$ | $2.7 \pm 0.5$ | $2.8 \pm 0.4$ | $2.7 \pm 0.5$ | $2.7 \pm 0.4$ |
| Count | 20 | 26 | 25 | 6 | 7 | - |

## J   Detailed Results Breakdowns

### J.1   RM-Bench & Reward Bench

Additional results presented in Table 12 and Table 13 are derived from the findings reported in Chen et al. (2025b).

### J.2   BBH Binary & MMLU-STEM Binary

Table 14 reports additional results from our BBH Binary and MMLU-STEM Binary.

### J.3   XSUM and FeedbackBench

Table 15 reports additional results, reproduced from prior work including Jia et al. (2023), to provide broader context and facilitate direct comparison across XSUM and FeedbackBench benchmarks.

Table 12: Comparison of existing models with R3 on RM-Bench. **Bolded numbers** indicate the best-performing results within each group section independently.

| Model | Domain | | | | Difficulty | | | Overall |
|---|---|---|---|---|---|---|---|---|
| | Chat | Math | Code | Safety | Easy | Medium | Hard | Avg. |
| Scalar RMs | | | | | | | | |
| steerlm-70b | 56.4 | 53.0 | 49.3 | 51.2 | 48.3 | 54.9 | 54.3 | 52.5 |
| tulu-v2.5-70b-preference-mix-rm | 58.2 | 51.4 | 55.5 | 87.1 | 72.8 | 65.6 | 50.7 | 63.0 |
| Mistral-7B-instruct-Unified-Feedback | 56.5 | 58.0 | 51.7 | 86.8 | 87.1 | 67.3 | 35.3 | 63.2 |
| RM-Mistral-7B | 57.4 | 57.0 | 52.7 | 87.2 | 88.6 | 67.1 | 34.9 | 63.5 |
| Eurus-RM-7b | 59.9 | 60.2 | 56.9 | 86.5 | 87.2 | 70.2 | 40.2 | 65.9 |
| internlm2-7b-reward | 61.7 | **71.4** | 49.7 | 85.5 | 85.4 | 70.7 | 45.1 | 67.1 |
| Skywork-Reward-Gemma-2-27B | 69.5 | 54.7 | 53.2 | 91.9 | 78.0 | 69.2 | 54.9 | 67.3 |
| ArmoRM-Llama3-8B-v0.1 | 67.8 | 57.5 | 53.1 | 92.4 | 82.2 | 71.0 | 49.8 | 67.7 |
| GRM-llama3-8B-sftreg | 62.7 | 62.5 | 57.8 | 90.0 | 83.5 | 72.7 | 48.6 | 68.2 |
| internlm2-20b-reward | 63.1 | 66.8 | 56.7 | 86.5 | 82.6 | 71.6 | 50.7 | 68.3 |
| Llama-3-OffsetBias-RM-8B | **71.3** | 61.9 | 53.2 | 89.6 | 84.6 | 72.2 | 50.2 | 69.0 |
| Nemotron-340B-Reward | 71.2 | 59.8 | **59.4** | 87.5 | 81.0 | 71.4 | **56.1** | 69.5 |
| URM-Llama-3.1-8B | 71.2 | 61.8 | 54.1 | 93.1 | 84.0 | 73.2 | 53.0 | 70.0 |
| Skywork-Reward-Llama-3.1-8B | 69.5 | 60.6 | 54.5 | **95.7** | 89.0 | 74.7 | 46.6 | 70.1 |
| infly/INF-ORM-Llama3.1-70B | 66.3 | 65.6 | 56.8 | 94.8 | **91.8** | **76.1** | 44.8 | **70.9** |
| Generative RMs | | | | | | | | |
| tulu-v2.5-dpo-13b-chatbot-arena-2023 | 64.9 | 52.3 | 50.5 | 62.3 | 82.8 | 60.2 | 29.5 | 57.5 |
| tulu-v2.5-dpo-13b-nectar-60k | 56.3 | 52.4 | 52.6 | 73.8 | 86.7 | 64.3 | 25.4 | 58.8 |
| stablelm-2-12b-chat | 67.2 | 54.9 | 51.6 | 65.2 | 69.1 | 63.5 | 46.6 | 59.7 |
| tulu-v2.5-dpo-13b-stackexchange-60k | 66.4 | 49.9 | 54.2 | 69.0 | 79.5 | 63.0 | 37.2 | 59.9 |
| Nous-Hermes-2-Mistral-7B-DPO | 58.8 | 55.6 | 51.3 | 73.9 | 69.5 | 61.1 | 49.1 | 59.9 |
| Claude-3-5-sonnet-20240620 | 62.5 | 52.6 | 54.5 | 64.4 | 73.8 | 63.4 | 45.9 | 61.0 |
| tulu-v2.5-dpo-13b-hh-rlhf-60k | 68.4 | 51.1 | 52.3 | 76.5 | 53.6 | 63.0 | 69.6 | 62.1 |
| tulu-2-dpo-13b | 66.4 | 51.4 | 51.8 | 85.4 | 86.9 | 66.7 | 37.7 | 63.8 |
| SOLAR-10.7B-Instruct-v1.0 | 78.6 | 52.3 | 49.6 | 78.9 | 57.5 | 67.6 | 69.4 | 64.8 |
| Llama3.1-70B-Instruct | 64.3 | 67.3 | 47.5 | 83.0 | 74.7 | 67.8 | 54.1 | 65.5 |
| Skywork-Critic-Llama-3.1-70B | 71.4 | 64.6 | 56.8 | 94.8 | 85.6 | 73.7 | 56.5 | 71.9 |
| GPT-4o-0806 | 67.2 | 67.5 | 63.6 | 91.7 | 83.4 | 75.6 | 58.7 | 72.5 |
| Gemini-1.5-pro | 71.6 | 73.9 | 63.7 | 91.3 | 83.1 | 77.6 | 64.7 | 75.2 |
| Prometheus-7B-v2.0 | 46.0 | 52.6 | 47.6 | 73.9 | 68.8 | 54.9 | 41.3 | 55.0 |
| JudgeLRM | 59.9 | 59.9 | 51.9 | 87.3 | 73.2 | 76.6 | 54.8 | 64.7 |
| RM-R1-Qwen-Instruct-7B | 66.6 | 67.0 | 54.6 | 92.6 | 79.2 | 71.7 | 59.7 | 70.2 |
| RM-R1-DeepSeek-Distilled-Qwen-7B | 64.0 | 83.9 | 56.2 | 85.3 | 75.9 | 73.1 | 68.1 | 72.4 |
| RM-R1-Qwen-Instruct-14B | **75.6** | 75.4 | 60.6 | 93.6 | 82.6 | 77.5 | 68.8 | 76.1 |
| RM-R1-Qwen-Instruct-32B | 75.3 | 80.2 | 66.8 | 93.9 | 86.3 | 80.5 | 70.4 | 79.1 |
| RM-R1-DeepSeek-Distilled-Qwen-14B | 71.8 | 90.5 | 69.5 | 94.1 | 86.2 | 83.6 | 74.4 | 81.5 |
| RM-R1-DeepSeek-Distilled-Qwen-32B | 74.2 | **91.8** | **74.1** | **95.4** | **89.5** | **85.4** | **76.7** | **83.9** |
| R3 Models (Ours) | | | | | | | | |
| R3-Qwen3-4B-LoRA-4k | 68.2 | 93.4 | 72.6 | 85.4 | 87.4 | 81.3 | 71.1 | 79.9 |
| R3-Qwen3-4B-LoRA-14k | 66.9 | 92.2 | 72.7 | 86.5 | 86.9 | 81.5 | 70.3 | 79.6 |
| R3-Qwen3-4B-4k | 68.9 | 92.3 | 72.5 | 86.5 | 86.5 | 81.4 | 72.3 | 80.0 |
| R3-Qwen3-4B-14k | 67.9 | 93.0 | 74.7 | 86.9 | 88.8 | 81.9 | 71.1 | 80.6 |
| R3-Qwen3-8B-LoRA-4k | 68.9 | 93.5 | 75.2 | 88.1 | 88.2 | 83.8 | 72.4 | 81.4 |
| R3-Qwen3-8B-LoRA-14k | 68.9 | 92.9 | 75.0 | 88.9 | 89.0 | 83.2 | 72.1 | 81.4 |
| R3-Qwen3-8B-4k | 70.8 | 92.9 | 74.2 | 89.2 | 87.9 | 83.4 | 74.0 | 81.8 |
| R3-Qwen3-8B-14k | 69.1 | 93.2 | 75.9 | 87.6 | 89.0 | 83.4 | 71.9 | 81.4 |
| R3-Qwen3-14B-LoRA-4k | 74.6 | 93.9 | 78.7 | **89.8** | 90.2 | 86.3 | **76.2** | **84.2** |
| R3-Qwen3-14B-LoRA-14k | 73.8 | 93.6 | 77.4 | 89.0 | 89.7 | 85.9 | 74.8 | 83.5 |
| R3-Qwen3-14B-4k | 74.0 | 93.7 | 77.2 | 89.3 | 89.7 | 85.3 | 75.6 | 83.6 |
| R3-Qwen3-14B-14k | 73.4 | 93.8 | **79.1** | 89.5 | **90.3** | **86.6** | 74.9 | 84.0 |
| R3-Phi-4-R+-14B-LoRA-4k | 71.4 | **94.4** | 78.2 | 86.2 | 88.7 | 84.3 | 74.7 | 82.5 |
| R3-Phi-4-R+-14B-LoRA-14k | 73.2 | 90.9 | 73.7 | 85.3 | 87.7 | 82.9 | 71.7 | 80.8 |
| R3-Phi-4-R+-14B-4k | **74.9** | 90.7 | 74.1 | 86.6 | 87.9 | 83.3 | 73.5 | 81.6 |
| R3-Phi-4-R+-14B-14k | 74.5 | 93.0 | 77.5 | 84.8 | 89.3 | 84.7 | 73.3 | 82.5 |
| R3-Qwen2.5-7B-LoRA-4K | 59.6 | 60.2 | 49.4 | 76.3 | 71.2 | 63.1 | 49.8 | 61.4 |
| R3-Qwen2.5-7B-4K | 69.6 | 75.5 | 59.8 | 86.9 | 80.2 | 74.2 | 64.5 | 73.0 |
| R3-Qwen2.5-7B-14K | 66.8 | 82.0 | 65.0 | 87.0 | 83.8 | 76.8 | 64.9 | 75.2 |
| R3-DeepSeek-Distilled-Qwen-14B-LoRA-4K | 69.0 | 90.3 | 70.5 | 85.8 | 85.9 | 81.6 | 69.3 | 78.9 |
| R3-DeepSeek-Distilled-Qwen-14B-LoRA-14K | 68.0 | 90.8 | 71.2 | 86.7 | 87.0 | 81.8 | 68.9 | 79.2 |
| R3-DeepSeek-Distilled-Qwen-14B-4K | 73.0 | 92.2 | 77.1 | 86.3 | 88.5 | 84.1 | 73.9 | 82.1 |
| R3-DeepSeek-Distilled-Qwen-14B-14K | 71.7 | 93.0 | 78.4 | 86.4 | 89.3 | 84.7 | 73.1 | 82.4 |
| Proprietary Models | | | | | | | | |
| GPT-4.1 mini | 67.6 | 73.0 | 71.3 | 90.7 | 87.0 | 78.4 | 61.7 | 75.7 |
| GPT-o4 mini | 77.6 | **93.0** | 80.8 | **93.4** | **92.0** | **88.7** | **78.0** | **86.2** |
| GPT-5 mini | **88.0** | 92.9 | **91.1** | 78.0 | 77.4 | 85.8 | 96.4 | 92.4 |
| DeepSeek-R1 | 78.6 | 66.2 | 81.9 | 88.7 | 86.9 | 82.2 | 67.3 | 78.8 |

Table 13: Comparison of existing models with R3 on RewardBench using pair-wise scoring. **Bolded numbers** indicate the best-performing results within each group section independently.

| Models | Chat | Chat Hard | Safety | Reasoning | Avg. |
|---|---|---|---|---|---|
| Scalar RMs | | | | | |
| Eurus-RM-7b | 98.0 | 65.6 | 81.4 | 86.3 | 82.8 |
| Internlm2-7b-reward | **99.2** | 69.5 | 87.2 | 94.5 | 87.6 |
| SteerLM-RM 70B | 91.3 | 80.3 | 92.8 | 90.6 | 88.8 |
| Cohere-0514 | 96.4 | 71.3 | 92.3 | 97.7 | 89.4 |
| Internlm2-20b-reward | 98.9 | 76.5 | 89.5 | 95.8 | 90.2 |
| ArmoRM-Llama3-8B-v0.1 | 96.9 | 76.8 | 90.5 | 97.3 | 90.4 |
| Nemotrom-4-340B-Reward | 95.8 | 87.1 | 91.5 | 93.6 | 92.0 |
| Skywork-Reward-Llama-3.1-8B | 95.8 | 87.3 | 90.8 | 96.2 | 92.5 |
| Skywork-Reward-Gemma-2-27B | 95.8 | **91.4** | 91.9 | 96.1 | 93.8 |
| infly/INF-ORM-Llama3.1-70B | 96.6 | 91.0 | **93.6** | **99.1** | **95.1** |
| Generative RMs | | | | | |
| Llama3.1-8B-Instruct | 85.5 | 48.5 | 75.6 | 72.1 | 70.4 |
| Llama3.1-70B-Intruct | **97.2** | 70.2 | 82.8 | 86.0 | 84.0 |
| Llama3.1-405B-Intruct | **97.2** | 74.6 | 77.6 | 87.1 | 84.1 |
| Claude-3-5-sonnet-20240620 | 96.4 | 74.0 | 81.6 | 84.7 | 84.2 |
| GPT-4o-0806 | 96.1 | 76.1 | 86.6 | 88.1 | 86.7 |
| Gemini-1.5-pro | 92.3 | 80.6 | 87.9 | 92.0 | 88.2 |
| Self-taught-evaluator-llama3.1-70B | 96.9 | 85.1 | 89.6 | 88.4 | 90.0 |
| SFR-LLaMa-3.1-70B-Judge-r | 96.9 | 84.8 | 91.6 | **97.6** | 92.7 |
| Skywork-Critic-Llama-3.1-70B | 96.6 | **87.9** | **93.1** | 95.5 | **93.3** |
| Prometheus-7B-v2.0 | 90.2 | 45.6 | 75.8 | 74.6 | 71.6 |
| m-Prometheus-14B | 93.6 | 59.0 | 85.1 | 84.8 | 80.6 |
| JudgeLRM | 92.9 | 56.4 | 78.2 | 73.6 | 75.2 |
| SynRM | 38.0 | 82.5 | 74.1 | 87.1 | 70.4 |
| RM-R1-DeepSeek-Distilled-Qwen-7B | 88.9 | 66.2 | 78.4 | 87.0 | 80.1 |
| RM-R1-Qwen-Instruct-7B | 94.1 | 74.6 | 85.2 | 86.7 | 85.2 |
| RM-R1-Qwen-Instruct-14B | 93.6 | 80.5 | 86.9 | 92.0 | 88.2 |
| RM-R1-DeepSeek-Distilled-Qwen-14B | 91.3 | 79.4 | 89.3 | 95.5 | 88.9 |
| R3 Models (Ours) | | | | | |
| R3-Qwen3-4B-LoRA-4K | 91.1 | 74.4 | 85.6 | 95.5 | 86.7 |
| R3-Qwen3-4B-LoRA-14K | 90.4 | 75.2 | 85.7 | 96.1 | 86.9 |
| R3-Qwen3-4B-4K | 88.3 | 77.4 | 86.1 | 95.3 | 86.8 |
| R3-Qwen3-4B-14K | 92.4 | 76.0 | 85.8 | 95.7 | 87.5 |
| R3-Qwen3-8B-LoRA-4K | 93.2 | 76.6 | 87.0 | 96.3 | 88.3 |
| R3-Qwen3-8B-LoRA-14K | 93.0 | 76.2 | 87.6 | 96.4 | 88.3 |
| R3-Qwen3-8B-4K | 91.6 | 79.8 | 87.7 | 95.8 | 88.7 |
| R3-Qwen3-8B-14K | 93.8 | 78.6 | 86.3 | 96.7 | 88.8 |
| R3-Qwen3-14B-LoRA-4K | 93.6 | **85.1** | **88.7** | 96.8 | **91.0** |
| R3-Qwen3-14B-LoRA-14K | 92.9 | 82.8 | 88.2 | **96.9** | 90.2 |
| R3-Qwen3-14B-4K | 92.6 | 81.0 | 88.4 | 96.6 | 89.7 |
| R3-Qwen3-14B-14K | 93.3 | 79.7 | 88.4 | **96.9** | 89.6 |
| R3-Phi-4-R$^+$-14B-LoRA-4K | 90.6 | 76.5 | 86.8 | 96.5 | 87.6 |
| R3-Phi-4-R$^+$-14B-LoRA-14K | 93.4 | 79.1 | 85.2 | 94.3 | 88.0 |
| R3-Phi-4-R$^+$-14B-4K | 92.6 | 79.0 | 85.8 | 96.3 | 88.4 |
| R3-Phi-4-R$^+$-14B-14K | **94.5** | 78.0 | 86.6 | 96.5 | 88.9 |
| R3-Qwen2.5-7B-LoRA-4K | 83.1 | 67.0 | 79.4 | 73.2 | 75.7 |
| R3-Qwen2.5-7B-4K | 85.9 | 75.3 | 85.5 | 85.1 | 82.9 |
| R3-Qwen2.5-7B-14K | 91.4 | 73.8 | 85.1 | 90.6 | 85.2 |
| R3-DeepSeek-Distilled-Qwen-14B-LoRA-4K | 90.8 | 75.6 | 84.6 | 93.1 | 86.0 |
| R3-DeepSeek-Distilled-Qwen-14B-LoRA-14K | 92.4 | 75.2 | 84.7 | 93.8 | 86.5 |
| R3-DeepSeek-Distilled-Qwen-14B-4K | 89.7 | 78.7 | 86.0 | 95.5 | 87.5 |
| R3-DeepSeek-Distilled-Qwen-14B-14K | 92.3 | 77.8 | 86.8 | 95.6 | 88.1 |
| Propretiary Models | | | | | |
| GPT-4.1 mini | **96.1** | 75.2 | 87.0 | 89.6 | 87.0 |
| GPT-o4 mini | 95.3 | **81.8** | 91.6 | **98.4** | **91.8** |
| GPT-5 mini | 95.3 | 81.6 | **92.0** | **98.4** | **91.8** |
| DeepSeek-R1 | 93.6 | 79.2 | 86.9 | 97.4 | 89.3 |

Table 14: Comparison of existing models with R3 on BBH & MMLU-STEM binary. **Bolded numbers** indicate the best-performing results between R3 models and baseline models. Proprietary models are bolded and compared independently.

| Models | BBH Binary Acc. | MMLU-STEM Acc. |
|---|---|---|
| Prometheus-7B-v2.0 | 54.0 | 56.5 |
| Selene-1-Mini-Llama-3.1-8B | 58.2 | 65.2 |
| RISE-Judge-Qwen2.5-7B | 63.1 | 76.9 |
| RISE-Judge-Qwen2.5-32B | **82.8** | **89.4** |
| R3 Models (Ours) | | |
| R3-Qwen3-4B-LoRA-4K | 89.0 | 92.1 |
| R3-Qwen3-4B-LoRA-14K | 88.9 | 92.2 |
| R3-Qwen3-4B-4K | 88.8 | 91.8 |
| R3-Qwen3-4B-14K | 89.3 | 92.0 |
| R3-Qwen3-8B-LoRA-4K | 90.8 | 93.5 |
| R3-Qwen3-8B-LoRA-14K | 90.8 | 93.6 |
| R3-Qwen3-8B-4K | 90.7 | 93.3 |
| R3-Qwen3-8B-14K | 90.7 | 93.6 |
| R3-Qwen3-14B-LoRA-4K | 91.7 | **94.8** |
| R3-Qwen3-14B-LoRA-14K | 91.9 | 94.5 |
| R3-Qwen3-14B-4K | 92.1 | 94.6 |
| R3-Qwen3-14B-14K | 92.1 | **94.8** |
| R3-Phi-4-R$^+$-14B-LoRA-4K | 91.4 | 93.3 |
| R3-Phi-4-R$^+$-14B-LoRA-14K | 91.3 | 93.5 |
| R3-Phi-4-R$^+$-14B-4K | 91.2 | 93.6 |
| R3-Phi-4-R$^+$-14B-14K | **92.2** | 94.4 |
| R3-Qwen2.5-7B-LoRA-4K | 71.7 | 81.8 |
| R3-Qwen2.5-7B-4K | 79.8 | 86.4 |
| R3-Qwen2.5-7B-14K | 81.1 | 88.3 |
| R3-DeepSeek-Distilled-Qwen-14B-LoRA-4K | 89.9 | 91.9 |
| R3-DeepSeek-Distilled-Qwen-14B-LoRA-14K | 90.0 | 92.2 |
| R3-DeepSeek-Distilled-Qwen-14B-4K | 91.3 | 92.9 |
| R3-DeepSeek-Distilled-Qwen-14B-14K | 91.1 | 93.0 |
| Propretiary Models | | |
| GPT-4.1 mini | 91.0 | 93.3 |
| GPT-o4 mini | 93.2 | 95.3 |
| GPT-5 mini | **95.0** | **96.5** |
| DeepSeek-R1 | 94.0 | 96.2 |

Table 15: Comparison of existing models with R3 on XSUM and FeedbackBench. **Bolded numbers** indicate the best-performing results between R3 models and baseline models. Proprietary models are bolded and compared independently.

| Models | XSUM | | | FeedbackBench |
| | Acc. | Kendall Tau | | Kendall Tau |
| | Faithfulness | Coherence | Relevance | |
|---|---|---|---|---|
| Llama-7B | 51.7 | - | - | - |
| Vicuna-7B | **55.5** | - | - | - |
| Alpaca-7B | 51.1 | - | - | - |
| UniEval | **84.3** | 0.07 | 0.03 | - |
| Prometheus-7B-v2.0 | 60.7 | 0.12 | 0.16 | **0.79** |
| Selene-1-Mini-Llama-3.1-8B | 56.4 | 0.16 | 0.36 | 0.78 |
| RISE-Judge-Qwen2.5-7B | 66.4 | 0.29 | 0.32 | 0.68 |
| RISE-Judge-Qwen2.5-32B | 71.0 | **0.30** | **0.39** | 0.74 |
| R3 Models (Ours) | | | | |
| R3-Qwen3-4B-LoRA-4K | **70.8** | 0.12 | 0.26 | 0.63 |
| R3-Qwen3-4B-LoRA-14K | 70.7 | 0.12 | 0.26 | 0.64 |
| R3-Qwen3-4B-4K | 66.8 | 0.23 | 0.27 | 0.63 |
| R3-Qwen3-4B-14K | 66.7 | 0.25 | 0.31 | 0.63 |
| R3-Qwen3-8B-LoRA-4K | 67.7 | 0.22 | 0.32 | 0.65 |
| R3-Qwen3-8B-LoRA-14K | 69.6 | 0.24 | 0.31 | 0.67 |
| R3-Qwen3-8B-4K | 68.0 | 0.36 | 0.31 | 0.66 |
| R3-Qwen3-8B-14K | 65.8 | 0.37 | 0.32 | **0.71** |
| R3-Qwen3-14B-LoRA-4K | 67.8 | 0.26 | 0.35 | 0.64 |
| R3-Qwen3-14B-LoRA-14K | 69.2 | 0.24 | 0.34 | 0.65 |
| R3-Qwen3-14B-4K | 67.8 | 0.34 | 0.34 | 0.68 |
| R3-Qwen3-14B-14K | 68.5 | 0.33 | **0.36** | **0.71** |
| R3-Phi-4-R$^+$-14B-LoRA-4K | 64.8 | **0.45** | 0.31 | 0.69 |
| R3-Phi-4-R$^+$-14B-LoRA-14K | 61.8 | 0.40 | 0.30 | 0.68 |
| R3-Phi-4-R$^+$-14B-4K | 67.5 | 0.36 | 0.30 | 0.69 |
| R3-Phi-4-R$^+$-14B-14K | 67.3 | 0.35 | 0.34 | 0.67 |
| R3-Qwen2.5-7B-LoRA-4K | 52.8 | 0.14 | 0.21 | 0.48 |
| R3-Qwen2.5-7B-4K | 65.1 | 0.29 | 0.29 | 0.64 |
| R3-Qwen2.5-7B-14K | 67.5 | 0.33 | 0.34 | 0.69 |
| R3-DeepSeek-Distilled-Qwen-14B-LoRA-4K | 58.4 | 0.21 | 0.31 | 0.64 |
| R3-DeepSeek-Distilled-Qwen-14B-LoRA-14K | 59.9 | 0.37 | 0.32 | 0.66 |
| R3-DeepSeek-Distilled-Qwen-14B-4K | 61.9 | 0.39 | 0.31 | 0.69 |
| R3-DeepSeek-Distilled-Qwen-14B-14K | 64.3 | 0.40 | 0.34 | **0.71** |
| Proprietary Models | | | | |
| GPT-4.1 mini | **72.6** | 0.07 | 0.38 | 0.69 |
| GPT-o4 mini | 69.1 | 0.16 | 0.30 | 0.66 |
| GPT-5 mini | 68.7 | **0.42** | **0.39** | 0.62 |
| DeepSeek-R1 | 60.4 | 0.35 | 0.38 | **0.72** |

