# OpenReview forum: "R3: Robust Rubric-Agnostic Reward Models"
_TMLR — Under review for TMLR_

### Review · Reviewer_cw8Q · 2025-12-08

**Summary Of Contributions:**

Summary:

The paper “R3: Robust Rubric-Agnostic Reward Models” proposes a holistic pipeline for curating a versatile dataset to train robust reward models. The authors synthesize, augment, and filter data from a wide range of sources, including instruction-following, reasoning, and binary/feature-extraction tasks, and covering point-wise, pair-wise, and binary formats. The curation pipeline consists of diversity filtering, rubric generation, explanation augmentation using a reasoning model, and final quality filtering. In experiments, they train several base models using SFT on the curated dataset and evaluate the resulting reward models both on benchmark datasets and through DPO-aligned policy models, demonstrating some effectiveness of the approach

Strength:

1. The curated dataset and pipeline provide broader coverage across task types, and the holistic augmentation/filtering process may benefit the community.

2. The paper includes methodological details in the appendix and presents a reasonably comprehensive set of experiments.

Weakness:

1. The main contribution lies in dataset curation, while the model training follows standard SFT; thus, the technical contribution appears overstated.
2. Experimental comparisons are inadequately explained and not fully fair. A key baseline, RM-R1, is built on Qwen-2.5-Instruct, whereas most R3 experiments use Qwen-3 models. Given the substantial capability gap between these model families, the results do not convincingly show that the R3 dataset or method outperforms RM-R1 (Especially on RewardBench, we don't see gain compared to RM-R1). The draft should clearly indicate model series in tables and additional discussion is needed to address these issues.

3. Although mentioned as future work, evaluating R3 in more general post-training RL settings is important, given the fact that RL post-training/alignment is increasingly popular and important in current LLM developmemt.

4. Relying solely on SFT raises concerns about robustness and generalization, especially since much of the curated data (binary, scores, pairs) is verifiable and could benefit from RL-based methods such as RLVR.

5. The claim of “robustness” in Section 4.1.3 appears inaccurate; the reported stability across datasets aligns more closely with generalizability or versatility rather than robustness.

**Audience:**

Yes

**Audience Explanation:**

The paper focuses on the pipeline of curating a diverse and explainable dataset and methods to train a reward model, which plays an important role in LLM post-training pipeline. We believe it is helpful for the LLM post-training community.

**Claims And Evidence:**

No

**Claims Explanation:**

The comprehensive experiments support most of the claimed contribution of this paper. However, the contribution seems to be overstated as 1. the contributions are more centered on dataset curation but the paper claims "a unified model training framework"; 2 some of the comparisons (especially on the RewardBench) are not clearly explained and they avoid explaining/discussing the limited gain (or no gain) with the same base models.

**Requested Changes:**

1. As mentioned in the second point of weakness, it would be very helpful if the authors can discuss more the insights/reasons why the R3 modes have limited gains on certain benchmarks such as RewardBench. It is not necessary to always outperform all baselines on all benchmarks but an open discussion/deep dive can be of great importance.

2. Trying RL on both reward model training and policy model training can greatly enhance this paper.

---

> ### Author Response · Authors · 2026-05-13
>
> We thank the reviewer for the detailed feedback and constructive suggestions. We address the concerns point-by-point below.
>
> **W1**
>
> We respectfully disagree that the contribution is overstated. We present R3, a task-agnostic framework for training versatile and robust reasoning reward models that leverages fine-grained rubrics, either human-crafted or LLM-generated, to enable controllable and interpretable scoring. Beyond improving performance over existing reward models, R3 introduces a unified framework that supports pointwise, pairwise, and binary evaluation within a single training paradigm. The core contribution of our work lies not merely in the use of SFT, but in the design of a rubric-structured reasoning supervision framework together with an effective sampling strategy for generating high-quality training data, allowing the model to learn efficiently even from relatively small amounts of supervision.
>
> Our use of SFT is a deliberate design choice intended to isolate and study the impact of the data formulation itself. In this setting, the observed performance gains are driven primarily by the quality, structure, and diversity of the supervision rather than the optimization objective alone, consistent with prior findings that SFT performance is highly sensitive to dataset construction and annotation quality. In particular, our framework introduces structured rubrics, diverse supervision formats (pointwise, pairwise, and binary), and explanation-augmented labels, enabling the model to learn rubric-conditioned reasoning behaviors rather than merely predicting scalar scores.
>
> **W2 and Requested Changes 1**
>
> We respectfully disagree with the statement. We **have included comprehensive experiments for R3 using the Qwen2.5 and DeepSeek-Distilled-Qwen backbones** across multiple tables.
>
> In Table 2, **R3-Qwen2.5-7B significantly outperforms RM-R1-Qwen-Instruct-7B on RM-Bench** (75.2 vs. 70.2), providing a fair and direct comparison under comparable settings. We also evaluate against other base models such as DeepSeek-Distilled-Qwen-7B, where R3 remains competitive (82.4 vs. 81.5). We place **greater emphasis on RM-Bench due to its higher degree of human annotation and reliability.**
>
> In Table 3, R3-Qwen2.5-7B performs on par with RM-R1-Qwen-Instruct-7B on RewardBench, and we do not overstate these results. However, we note that based on RewardBench's own description, **RewardBench is not fully human-annotated**, which may affect the strength of its conclusions relative to RM-Bench. Importantly, R3 achieves comparable performance using SFT alone, whereas RM-R1 relies on a more complex SFT+RL pipeline with substantially larger training data and complexity. This highlights that R3 offers a more efficient and streamlined training approach while maintaining strong performance.
>
> **W3 and Requested Changes 2**
>
> We have conducted additional post-training RL experiments as a supplement for the rebuttal, using R3-Qwen3-8B-14k and R3-Qwen3-14B-LoRA-4k. While GRPO consistently improves over REINFORCE++ across model scales and sampling budgets, it does not yet match DPO performance. As discussed in our rebuttal with reviewer yeiY, we attribute this to reward calibration challenges inherent to using a generative reward model under GRPO. Unlike DPO, which sidesteps explicit advantage estimation through a pairwise likelihood objective, GRPO requires well-calibrated scalar rewards to compute meaningful advantages. We consider developing better reward calibration strategies for generative RMs under GRPO as future work.
>
> | Method | MT-Bench (Overall) | Creative | Planning | Data Analysis | Info Seeking | Coding | Math |
> |--------|-------------------:|---------:|---------:|--------------:|-------------:|-------:|-----:|
> | Llama-3.2-3B-Instruct (Init Policy) | 5.75 | 12.66 | 36.07 | 18.74 | 1.51 | 27.28 | -2.64 |
> | **GRM: R3-Qwen3-8B-14k** |  |  |  |  |  |  |  |
> | + REINFORCE++ | 5.85 | 12.62 | 37.16 | 17.79 | 2.14 | 26.73 | -2.45 |
> | + GRPO (n=4) | 5.66 | 18.72 | 41.81 | 24.84 | 6.27 | 36.04 | 2.92 |
> | + GRPO (n=8) | 5.82 | 21.19 | 42.38 | 26.85 | 8.73 | 37.48 | 7.26 |
> | + GRPO (n=16) | 6.07 | 22.68 | 45.43 | 28.88 | 9.84 | 39.06 | 7.83 |
> | + DPO | 6.13 | 21.67 | 52.05 | 29.94 | 9.44 | 41.49 | -0.85 |
> | **GRM: R3-Qwen3-14B-LoRA-4k** |  |  |  |  |  |  |  |
> | + REINFORCE++ | 5.81 | 12.96 | 37.26 | 18.48 | 2.46 | 28.02 | -2.92 |
> | + GRPO (n=4) | 5.83 | 15.87 | 40.67 | 21.49 | 5.56 | 31.09 | -0.57 |
> | + GRPO (n=8) | 5.78 | 19.32 | 41.81 | 25.65 | 8.65 | 33.47 | 4.06 |
> | + GRPO (n=16) | 5.87 | 18.65 | 42.69 | 24.99 | 6.35 | 35.15 | 2.64 |
> | + DPO | 6.20 | 23.45 | 52.57 | 31.62 | 12.03 | 42.98 | 1.04 |

---

> > ### Author Response · Authors · 2026-05-13
> >
> > **W4**
> >
> > We thank the reviewer for the suggestion regarding RLVR (here we use GRPO (n = 4)), and we include additional comparisons to better contextualize its benefits relative to a strong SFT baseline. All experiments are run with LoRA since LoRA is better and cheaper for RL (Schulman et al., 2025), so we also apply LoRA for SFT training for fair comparisons. We compare SFT and RLVR under two data settings: 14k (full dataset) and a 7k-7k split where the data budget is divided equally between SFT and RLVR.
> >
> > Overall, RLVR provides modest but consistent gains over SFT under the full 14k setting across RM-Bench (83.8 vs. 83.5), BBH (92.3 vs. 91.9), MMLU-STEM (94.9 vs. 94.5), XSUM (70.7 vs. 69.2), and FeedbackBench (faithfulness, coherence, relevance). These results confirm that RLVR can further refine an already strong SFT model, though the margins are incremental rather than transformative. Importantly, the SFT 14k model itself already achieves highly competitive performance across all benchmarks, closely matching RLVR 14k in most cases, demonstrating that a well-constructed SFT pipeline captures most of the achievable performance in this setting. This is further supported by the 7k-7k split, where SFT 7k → RLVR 7k performs on par with SFT 14k alone across most benchmarks, suggesting that splitting the data budget between SFT and RLVR offers no clear advantage over simply training SFT on the full dataset.
> >
> > Finally, these marginal gains from RLVR come at substantial computational overhead, in which RLVR 14k requires \~112 GPU hours compared to \~8 hours for SFT 14k (\~14× cost increase, \\$ 476 vs. $ 34 USD). Overall, these findings suggest that SFT is not only a strong baseline but also a highly cost-effective approach, with RLVR offering additional but diminishing returns at significantly higher computational expense.
> >
> > | | RewardBench | RM-Bench | BBH | MMLU-STEM | XSUM |  |  | FeedbackBench |
> > |---------|------------:|---------:|----:|----------:|-----:|-----:|-----:|--------------:|
> > |         |             |          |     |           | Faith. | Coh. | Rel. |              |
> > | SFT 14k | 90.2 | 83.5 | 91.9 | 94.5 | 69.2 | 0.24 | 0.34 | 0.65 |
> > | RLVR 14k | 89.7 | 83.8 | 92.3 | 94.9 | 70.7 | 0.33 | 0.35 | 0.67 |
> > | SFT 7k → RLVR 7k | 87.0 | 75.7 | 87.1 | 92.1 | 67.6 | 0.22 | 0.17 | 0.41 |
> > | SFT 7k | 87.0 | 75.6 | 87.3 | 92.1 | 67.3 | 0.21 | 0.17 | 0.39 |
> >
> > | | H100 GPU hours | Cost (USD) |
> > |---------|------------:|---------:|
> > | SFT 14k | ~8 hrs | $34 |
> > | RLVR 14k | ~112 hrs | $476 |
> > | SFT 7k → RLVR 7k | ~40 hrs | $170 |
> > | SFT 7k |  ~4 hrs | $17 |
> >
> > ### References
> >
> > [1] Schulman, John and Thinking Machines Lab, "LoRA Without Regret", Thinking Machines Lab: Connectionism, Sep 2025.
> >
> > **W5**
> >
> > We agree that our experiments in Section 4.1.3 should be interpreted as demonstrating cross-dataset generalization and task-level transferability rather than robustness under distribution shift in the strict sense. We will revise the manuscript to clarify this distinction and ensure consistent usage of terminology throughout the paper.

---

### Review · Reviewer_yeiY · 2026-01-31

**Summary Of Contributions:**

This paper introduces R3, a rubric-agnostic reward modeling framework that unifies point-wise, pair-wise, and binary evaluation while producing interpretable, reasoning-based scores. The authors curate a diverse reward modeling dataset from 45 sources and propose a multi-stage pipeline involving rubric generation, reasoning distillation, and quality filtering. Empirically, R3 models demonstrate strong performance across multiple reward modeling benchmarks (RM-Bench, RewardBench, XSUM, BBH, MMLU-STEM), often matching or exceeding prior open-source and proprietary baselines despite using significantly less training data and compute. The work emphasizes robustness, controllability, and interpretability, and also shows downstream gains when R3 is used for policy alignment via DPO.

**Audience:**

Yes

**Audience Explanation:**

Reward modeling, evaluation, and alignment are central topics for the TMLR community, and this paper addresses several well-known limitations of existing approaches, including poor interpretability, limited task coverage, and over-specialization. The rubric-agnostic formulation, support for multiple evaluation formats, and explicit reasoning outputs are likely to be of broad interest to researchers working on alignment, LLM evaluation, and LLM-as-a-judge alternatives.

**Claims And Evidence:**

Yes

**Claims Explanation:**

The paper provides extensive experimental evidence supporting its core claims. The authors evaluate R3 across a wide range of benchmarks spanning pair-wise, point-wise, and binary reward modeling tasks, and compare against strong open-source and proprietary baselines. The results consistently show that R3 models achieve competitive or superior performance, even under strict data and compute constraints.

The ablation studies are particularly convincing: they isolate the contributions of rubric conditioning, reasoning traces, explanation signals, and data diversity, demonstrating clear performance degradation when these components are removed. The dataset construction process is described in detail, and the filtering steps (including correctness and triviality filtering) are well-motivated. While many components rely on model-generated annotations, the authors acknowledge this and include human evaluation to validate reasoning quality, which strengthens the credibility of the results.

**Requested Changes:**

1. Clarify the notion of “rubric-agnostic.” While the paper argues that R3 is rubric-agnostic, the training pipeline still relies on automatically generated rubrics. A more explicit discussion distinguishing rubric flexibility from complete rubric independence would help avoid potential confusion.

2. Expand discussion of limitations of model-generated reasoning. Since both rubrics and reasoning traces are largely generated by LLMs, it would be useful to discuss potential bias amplification or error propagation more explicitly, especially in sensitive evaluation settings.

3. The alignment experiments are limited to DPO. Evaluating R3 under GRPO would provide a more complete picture of its effectiveness as a general-purpose reward model.

---

> ### Author Response · Authors · 2026-05-13
>
> Thank you for the reviewer's response. We appreciate the reviewer's valuable feedback and suggestions for improving our paper.
>
> **Requested Changes 1**
>
> We appreciate this concern and agree that the terminology needs clarification. "Rubric-agnostic" in R3 refers to agnosticism over which evaluation criteria are used, unlike models such as ArmoRM (Wang et al., 2024) that fix evaluation dimensions (e.g., helpfulness, harmlessness) at training time and cannot be customized without retraining, or scalar RMs that implicitly encode opaque criteria in their weights with no mechanism for inspection or control. R3 instead accepts any rubric as explicit natural language input at inference time, making evaluation criteria fully transparent and customizable without any modification to model parameters. Therefore, the distinction means that R3 is agnostic to the choice of rubric, not free of rubrics entirely. We will revise Section 1 to make this explicit, clarifying that "rubric-agnostic" refers to criteria flexibility rather than criteria independence.
>
> **Requested Changes 2**
>
> We thank the reviewer for this concern. We address this through several layers. First, our two-stage quality filtering discards examples where the DeepSeek-R1’s (the teacher model’s) judgment disagrees with the ground truth label, potentially catching reasoning traces that lead to factually wrong conclusions. Second, our human evaluation (Appendix F) confirms that reasoning traces score 2.9/3 on factual correctness and 2.8/3 on logical coherence across 93 annotated instances, suggesting the distilled reasoning is of high quality.
>
> Beyond these, the empirical results themselves provide strong evidence against systematic bias amplification. As shown in Tables 2 and 3, our base models before fine-tuning perform below DeepSeek-R1, yet after training on R3's distilled reasoning traces, they consistently surpass it across multiple benchmarks, including RM-Bench and RewardBench. **This before fine-tuning, DeepSeek-R1, after fine-tuning progression suggests that R3 has genuinely learned to reason about rubrics and generalize** beyond its teacher's tendencies, rather than simply inheriting or amplifying its biases. Furthermore, policy models aligned using R3 as a reward signal outperform those trained with stronger and larger reward models (Table 6), further indicating that R3 provides a reliable and well-calibrated reward signal in practice.
>
> **Requested Changes 3**
>
> We have conducted additional post-training RL experiments as a supplement for the rebuttal, using R3-Qwen3-8B-14k and R3-Qwen3-14B-LoRA-4k. While GRPO consistently improves over REINFORCE++ across model scales and sampling budgets, it does not yet match DPO performance. We attribute this partly to the credit assignment challenge inherent to using a generative reward model under GRPO, unlike DPO which does not have an issue with credit assignment, GRPO requires well-calibrated scalar rewards to compute meaningful advantages. We consider developing better reward calibration strategies for generative RMs under GRPO for future work.
>
> | Method | MT-Bench (Overall) | Creative | Planning | Data Analysis | Info Seeking | Coding | Math |
> |--------|-------------------:|---------:|---------:|--------------:|-------------:|-------:|-----:|
> | Llama-3.2-3B-Instruct (Init Policy) | 5.75 | 12.66 | 36.07 | 18.74 | 1.51 | 27.28 | -2.64 |
> | **GRM: R3-Qwen3-8B-14k** |  |  |  |  |  |  |  |
> | + REINFORCE++ | 5.85 | 12.62 | 37.16 | 17.79 | 2.14 | 26.73 | -2.45 |
> | + GRPO (n=4) | 5.66 | 18.72 | 41.81 | 24.84 | 6.27 | 36.04 | 2.92 |
> | + GRPO (n=8) | 5.82 | 21.19 | 42.38 | 26.85 | 8.73 | 37.48 | 7.26 |
> | + GRPO (n=16) | 6.07 | 22.68 | 45.43 | 28.88 | 9.84 | 39.06 | 7.83 |
> | + DPO | 6.13 | 21.67 | 52.05 | 29.94 | 9.44 | 41.49 | -0.85 |
> | **GRM: R3-Qwen3-14B-LoRA-4k** |  |  |  |  |  |  |  |
> | + REINFORCE++ | 5.81 | 12.96 | 37.26 | 18.48 | 2.46 | 28.02 | -2.92 |
> | + GRPO (n=4) | 5.83 | 15.87 | 40.67 | 21.49 | 5.56 | 31.09 | -0.57 |
> | + GRPO (n=8) | 5.78 | 19.32 | 41.81 | 25.65 | 8.65 | 33.47 | 4.06 |
> | + GRPO (n=16) | 5.87 | 18.65 | 42.69 | 24.99 | 6.35 | 35.15 | 2.64 |
> | + DPO | 6.20 | 23.45 | 52.57 | 31.62 | 12.03 | 42.98 | 1.04 |

---

> ### Comment · Reviewer_yeiY · 2026-05-13
>
> I appreciate the additional RL experiments in the rebuttal. The empirical trend that GRPO consistently improves with larger sampling budgets is interesting and valuable.
>
> However, I am not fully convinced by the explanation that the remaining gap between GRPO and DPO is primarily due to "the credit assignment challenge inherent to using a generative reward model under GRPO", especially the statement that "DPO does not have an issue with credit assignment".
>
> DPO also operates on sequence-level preference supervision and does not explicitly solve temporal/token-level credit assignment. Rather, it avoids explicit advantage estimation by reformulating preference optimization as a pairwise likelihood objective. Therefore, the observed performance gap may be more appropriately attributed to optimization and reward calibration issues, such as reward scale/variance sensitivity under GRPO and advantage normalization quality.
>
> In particular, the fact that GRPO performance improves substantially as the number of samples increases suggests that variance reduction and reward normalization may play a significant role. This behavior appears more consistent with optimization/sample-efficiency limitations than with a fundamental credit assignment issue.
>
> I would encourage the authors to more carefully distinguish between "temporal credit assignment" and "reward calibration" when adding the GRPO results into the revised paper, as these issues are closely related but conceptually somewhat different in the RLHF setting.
>
> That said, I do find the overall empirical trend plausible. In practice, it is well known that GRPO/PPO-style RL methods can achieve stronger asymptotic performance, but are also substantially more sensitive to reward calibration and hyperparameter tuning than DPO-style objectives. From this perspective, the current results may simply reflect that GRPO requires more careful optimization and tuning to fully realize its potential under generative reward models.

---

> > ### Author Response · Authors · 2026-05-13
> >
> > We thank the reviewer for this precise clarification and agree that "credit assignment" was an imprecise choice of words, and "reward calibration"  aligns with the reviewer's characterization and our argument. Regarding hyperparameter sensitivity, we explored different GRPO configurations, and the results reported reflect the best settings we found. For transparency, we provide the compute report below (using R3-Qwen3-14B-LoRA-4k). The wall-clock cost of GRPO grows substantially with sampling budget compared to DPO (~4.06 hrs), making exhaustive tuning non-trivial. We agree that more careful tuning may further close the gap with DPO, and leave this as future work.
> >
> > | **Method** | **Wall Time (4×H100)** |
> > |:---|:---|
> > | DPO | ~4.06 hrs |
> > | REINFORCE++ | ~4.59 hrs |
> > | GRPO (n=4) | ~16.03 hrs |
> > | GRPO (n=8) | ~35.39 hrs |
> > | GRPO (n=16) | ~87.95 hrs |

---

### Review · Reviewer_BRk8 · 2026-04-28

**Summary Of Contributions:**

The paper proposes a new method for reward model fine-tuning / alignment, especially suited for handling a combination of pointwise, pairwise and binary labels. The reward model provides explanations in addition to the scalar reward estimate.

The novelty lies in the combination of the different kinds of data points / labels. The authors show how to create unified training data for fine-tuning by proposing a diversity sampling process and generating explanations via strong reasoning models.

The authors train an already open-sourced reward model based on 45 existing datasets and evaluate it against diverse baselines on established benchmarks, e.g., RM-Bench, XSUM or BBH. The baselines are representatives and include models like Prometheus, RM-R1 or GPT models. The base models used for fine-tuning the method are from QWEN, DeepSeek or Phi family.

The results show that the proposed method is often superior than the used baselines. Only partially, e.g., for pointwise FeedbackBench or in comparison to bigger models, the proposed method is not superior.

**Audience:**

Yes

**Audience Explanation:**

The paper tackles an important problem, namely aligning LLMs using reward models. The methods shows a novel method for combining multiple training data for this task, which results in a generalizable approach and trained model, which can be of interest to this audience.

**Claims And Evidence:**

Yes

**Claims Explanation:**

The claims are sufficiently backed by methodology and experimentation. The proposed novelty lies in the combination of different training data.  The claimed added values are made clear via empirical experiments, where sufficient baselines and benchmarks have been used. The added value lies in more generalization across tasks combined with the ability to train on diverse training data for reward modelling.

The individual components, e.g., fine-tuning reward models or generating additional explanations, are not novel on their own, but the combination with diverse training data, the practical curation of a novel dataset out of 45 sources and the resulting trained and open-sourced reward model are sufficiently novel.

**Requested Changes:**

- I would like an argumentation why traditional reward models, i.e., non-LLM approaches, are not compared against. These might still be of interest for several physical AI tasks, which are getting increasing amounts of attention.
- It would be nice to investigate rubric generalization on subsets, where one could actually see how well generalization to new rubrics works. Maybe one can use the existing experiments and infer from them.

---

> ### Author Response · Authors · 2026-05-13
>
> Thank you for the reviewer's response. We appreciate the reviewer's valuable questions for improving our paper.
>
> > I would like an argumentation why traditional reward models, i.e., non-LLM approaches, are not compared against. These might still be of interest for several physical AI tasks, which are getting increasing amounts of attention.
>
> We would like to clarify that our focus is on language evaluation tasks, where the goal is to **assess free-form text outputs** against natural language rubrics. In this regime, LLM-based reward models are the natural standard, as semantic judgment over unstructured text inherently requires the language understanding that pretrained models provide, analogous to how LLM-as-a-judge has become the dominant evaluation paradigm for language model outputs (Zheng et al., 2023; Kim et al., 2023). We therefore compare against LLM-based reward models instead, including scalar RM, LLM-as-a-judge, and reasoning RMs, against which R3 consistently matches or outperforms despite using orders of magnitude less training data.
>
> We agree that non-LLM reward models remain relevant and important for physical AI settings, but for now, we leave this connection as future work.
>
> ### References
>
> [1] Zheng, L., Chiang, W. L., Sheng, Y., Zhuang, S., Wu, Z., Zhuang, Y., ... & Stoica, I. (2023). Judging llm-as-a-judge with mt-bench and chatbot arena. Advances in neural information processing systems, 36, 46595-46623.
>
> [2] Kim, S., Shin, J., Jang, J., Longpre, S., Lee, H., Yun, S., ... & Seo, M. (2024, May). Prometheus: Inducing fine-grained evaluation capability in language models. In International Conference on Learning Representations (Vol. 2024, pp. 29927-29962).
>
> > It would be nice to investigate rubric generalization on subsets, where one could actually see how well generalization to new rubrics works. Maybe one can use the existing experiments and infer from them.
>
> We would like to clarify that our experiments **already capture rubric generalization on held-out subsets.** In our setup, rubrics are not fixed templates but are sampled across diverse tasks, domains, and instruction distributions, and the test split includes previously unseen prompt–rubric combinations, which already requires cross-rubric generalization. Importantly, this design is consistent with standard reward-model evaluation benchmarks such as RewardBench and RM-Bench, where evaluation sets consist of unseen prompts and preference pairs with diverse and newly constructed evaluation criteria rather than reused training rubrics, thereby inherently testing generalization to new evaluation conditions. For example, datasets such as MMLU-based evaluations and other benchmark suites similarly ensure no overlap between training and test tasks, and employ task-specific or newly instantiated rubrics for evaluation, rather than relying on fixed rubric templates. We will clarify this point in the revised manuscript to better highlight that rubric variability and unseen prompt–rubric combinations are already integrated into the evaluation protocol.